# More homogeneous capillary flow and oxygenation in deeper cortical layers correlate with increased oxygen extraction

**Baoqiang Li[1†], Tatiana V Esipova[2,3†], Ikbal Sencan[1], Kıvılcım Kılıç[4], Buyin Fu[1], Michele Desjardins[5], Mohammad Moeini[6,7], Sreekanth Kura[1], Mohammad A Yaseen[1], Frederic Lesage[6,7], Leif Østergaard[8], Anna Devor[1,4,5], David A Boas[1,9], Sergei A Vinogradov[2,3], Sava Sakadžić[1]\***

[1]Athinoula A. Martinos Center for Biomedical Imaging, Massachusetts General Hospital, Harvard Medical School, Charlestown, United States; [2]Department of Biochemistry and Biophysics, University of Pennsylvania, Philadelphia, United States; [3]Department of Chemistry, University of Pennsylvania, Philadelphia, United States; [4]Department of Neurosciences, University of California, San Diego, La Jolla, United States; [5]Department of Radiology, University of California, San Diego, La Jolla, United States; [6]Institute of Biomedical Engineering, École Polytechnique de Montréal, Montréal, Canada; [7]Research Centre, Montreal Heart Institute, Montréal, Canada; [8]Center of Functionally Integrative Neuroscience and MINDLab, Institute of Clinical Medicine, Aarhus University, Aarhus, Denmark; [9]Department of Biomedical Engineering, Boston University, Boston, United States

**\*For correspondence:**
sava.sakadzic@mgh.harvard.edu

[†]These authors contributed equally to this work

**Competing interests:** The authors declare that no competing interests exist.

**Abstract** Our understanding of how capillary blood flow and oxygen distribute across cortical layers to meet the local metabolic demand is incomplete. We addressed this question by using two-photon imaging of resting-state microvascular oxygen partial pressure ($PO_2$) and flow in the whisker barrel cortex in awake mice. Our measurements in layers I-V show that the capillary red-blood-cell flux and oxygenation heterogeneity, and the intracapillary resistance to oxygen delivery, all decrease with depth, reaching a minimum around layer IV, while the depth-dependent oxygen extraction fraction is increased in layer IV, where oxygen demand is presumably the highest. Our findings suggest that more homogeneous distribution of the physiological observables relevant to oxygen transport to tissue is an important part of the microvascular network adaptation to local brain metabolism. These results will inform the biophysical models of layer-specific cerebral oxygen delivery and consumption and improve our understanding of the diseases that affect cerebral microcirculation.
DOI: https://doi.org/10.7554/eLife.42299.001

## Introduction

Normal brain functioning is critically dependent on the adequate and uninterrupted supply of oxygen to brain tissue (*Attwell and Laughlin, 2001*; *Raichle and Gusnard, 2002*; *Raichle et al., 2001*). Significant efforts have been made over the years to investigate the regulation of cerebral blood flow (CBF), including auto-regulation of CBF in response to the changes in cerebral perfusion pressure (*Aaslid et al., 1989*; *Paulson et al., 1990*) and the processes underlying neurovascular coupling (*Anenberg et al., 2015*; *Cai et al., 2018*; *Girouard and Iadecola, 2006*; *Iordanova et al., 2015*;

*Vazquez et al., 2010*). Furthermore, neuronal and microvascular densities vary greatly between cortical layers (*Blinder et al., 2013*; *Sakadžić et al., 2014*; *Weber et al., 2008*; *Wu et al., 2016*), suggesting a laminar variation of tissue metabolism (*de Kock et al., 2007*; *Hyder et al., 2013*). However, it is still not well understood *how blood flow and oxygenation are distributed through the microvascular network to support an adequate tissue oxygenation across the closely spaced, but morphologically and metabolically heterogeneous cortical layers*. Answering this question is important to improve our understanding of the normal brain physiology as well as brain diseases that affect cerebral microcirculation (*Berthiaume et al., 2018*; *Iadecola, 2016*; *Iadecola, 2017*; *Pantoni, 2010*; *Zlokovic, 2011*).

The recent development of tools for in vivo microvascular oxygen imaging enabled investigation of brain oxygen delivery and consumption within the arteriolar, venular and capillary domains over large tissue volumes (*Cao et al., 2017*; *Chong et al., 2015a*; *Hu et al., 2009*; *Lecoq et al., 2011*; *Parpaleix et al., 2013*; *Sakadžić et al., 2010*; *Sakadžić et al., 2015*; *Wang et al., 2011*; *Yaseen et al., 2009*). In addition, distributions of microvascular blood flow and oxygen in mice at rest have been assessed in several studies using optical coherence tomography, two-photon microscopy and photoacoustic imaging (*Cao et al., 2017*; *Chong et al., 2015b*; *Gutiérrez-Jiménez et al., 2018*; *Lyons et al., 2016*; *Moeini et al., 2018*; *Sakadžić et al., 2011*; *Sakadžić et al., 2014*; *Santisakultarm et al., 2012*; *Santisakultarm et al., 2014*; *Srinivasan et al., 2015*). However, only in few previous studies blood flow properties at rest have been considered as a function of cortical depth, but without information about capillary oxygenation and branching order (*Gutiérrez-Jiménez et al., 2016*; *Kleinfeld et al., 1998*; *Li et al., 2016*; *Merkle and Srinivasan, 2016*). These studies in anesthetized mice showed that capillary red-blood-cell (RBC) flux and/or speed slightly decreased with cortical depth at rest, and that smaller transit time heterogeneity occurred in the deeper cortical layers. By using two-photon phosphorescence lifetime microscopy (2PLM) of oxygen (*Finikova et al., 2008*), distributions of capillary oxygen concentration were reported in both anesthetized (*Lecoq et al., 2011*; *Parpaleix et al., 2013*; *Sakadžić et al., 2014*) and awake (*Lyons et al., 2016*; *Moeini et al., 2018*) mice. The use of anesthesia in some of these studies could have significantly affected both brain CBF and metabolism (*Alkire et al., 1999*; *Goldberg et al., 1966*). Moreover, all earlier studies using 2PLM were limited by smaller sample size and insufficient imaging penetration depth due to limitations of the applied oxygen probe – PtP-C343 (*Finikova et al., 2008*). Lyons et al. for the first time measured the distributions of capillary blood flow and PO$_2$ as a function of cortical depth in the somatosensory cortex in awake mice (*Lyons et al., 2016*). However, the measurements were performed in less than 100 capillaries across n = 3 mice with the imaging depth of ≤410 μm. Due to these limitations, a detailed layer-specific analysis with the acquired data would be challenging. Moeini et al. conducted imaging study with 2PLM in awake mice through a thinned-skull cranial window (*Moeini et al., 2018*). This procedure was less invasive, but it further limited imaging depth.

In addition to the experimental observations, several recent studies using numerical modeling based on the realistic and artificial vascular anatomical networks (VANs) predicted layer-dependent blood flow (*Hartung et al., 2018*; *Schmid et al., 2017*) and oxygen distributions (*Gagnon et al., 2016*; *Gould and Linninger, 2015*; *Gould et al., 2017*; *Linninger et al., 2013*; *Lücker et al., 2018a*; *Lücker et al., 2018b*). Developing and using accurate biophysical models of oxygen advection and diffusion based on large-scale realistic VANs is a key component in our quest to better understand the regulation of microvascular blood flow. However, current numerical models need to be improved, and their predictions need to be validated by more comprehensive experimental measurements of the physiological observables involved in microvascular oxygen transport to tissue.

The development of 2PLM of oxygen also enabled measurements of erythrocyte-associated transients (EAT) in cortical capillaries (*Lecoq et al., 2011*). EAT were first theoretically predicted by *Hellums (1977)* and extensively investigated over the last four decades using analytical and numerical approaches (*Federspiel and Popel, 1986*; *Hellums, 1977*; *Lücker et al., 2015*; *Lücker et al., 2017*; *Popel, 1989*). Originally, they were experimentally observed in peripheral capillaries (*Barker et al., 2007*; *Golub and Pittman, 2005*), but the full confirmation within the more challenging three-dimensional cortical capillary network was made possible only recently with advent of 2PLM (*Lecoq et al., 2011*; *Parpaleix et al., 2013*). Since EAT is tightly related to the intravascular resistance to oxygen transport to tissue (*Golub and Pittman, 2005*; *Hellums, 1977*), their direct measurements are critical

for better understanding of the oxygen delivery through the capillary network. However, the dependence of EAT on cortical layer has not been fully explored.

To this end, in the present work, we applied 2PLM to measure the absolute intravascular $PO_2$ in a large number of arterioles, venules and capillaries, as well as RBC-$PO_2$, InterRBC-$PO_2$, EAT and RBC flow properties in capillaries as a function of cortical depth within the range of 0–600 μm. We used a new phosphorescent oxygen probe – Oxyphor2P (*Esipova et al., 2019*). Compared to its predecessor – PtP-C343 (*Finikova et al., 2008*), Oxyphor2P exhibits longer excitation and emission wavelength maxima, higher quantum yield and a larger two-photon absorption cross-section, facilitating simultaneous mapping of microvascular $PO_2$ and capillary RBC flux in cortical layers I-V in mice and making it possible to significantly increase both the sample size and imaging penetration depth in comparison to the previous studies (*Lecoq et al., 2011*; *Lyons et al., 2016*; *Moeini et al., 2018*; *Parpaleix et al., 2013*; *Sakadžić et al., 2010*; *Sakadžić et al., 2014*). Moreover, our measurements were performed in head-restrained awake mice, and thus were free of the confounding effects of anesthesia on neuronal activity, CBF and brain metabolism. With the results, we report that 1) in the whisker barrel cortex in awake mice the oxygen extraction fraction (OEF), measured at different cortical depths (i.e., depth-dependent OEF), reaches its maximum at the depth range of 320–450 μm, corresponding to cortical layer IV, where the neuron and capillary densities are the largest, and, presumably, oxygen consumption is the highest; and 2) the increased OEF is accompanied by the more homogenously distributed capillary $PO_2$ and RBC flux, as well as by a decrease in the intracapillary resistance to oxygen delivery (inferred from the EAT magnitude). These experimental results enabled quantification of parameters of importance for oxygen transport to tissue and put forward a potential mechanism, by which the microvascular networks at rest may adapt to the heterogeneous metabolic demands in different cortical layers. We anticipate that our results will improve our understanding of the normal brain function as well as of diseases that affect the cerebral microcirculation (*Girouard and Iadecola, 2006*; *Iadecola, 2016*; *Müller et al., 2017*; *Pantoni, 2010*; *Wardlaw et al., 2013*; *Zlokovic, 2011*). In addition, detailed knowledge of the distributions of capillary RBC flux and oxygenation at different depths should inform the next generation of biophysical models of the layer-specific blood flow, oxygen delivery and consumption (*Gagnon et al., 2016*; *Gould and Linninger, 2015*; *Gould et al., 2017*; *Guibert et al., 2010*; *Guibert et al., 2012*; *Hartung et al., 2018*; *Linninger et al., 2013*; *Lorthois and Lauwers, 2012*; *Lorthois et al., 2011*; *Peyrounette et al., 2018*; *Schmid et al., 2017*), as well as modeling of the impact of various RBC flow properties on oxygen delivery (*Lücker et al., 2018a*; *Lücker et al., 2018b*).

## Results

### Oxygen extraction fraction increases in the deeper cortical layers

We used a home-built two-photon microscope (*Sakadžić et al., 2010*; *Yaseen et al., 2015*) (*Figure 1a*) to measure the resting intravascular $PO_2$ in the whisker barrel cortex in head-restrained awake mice through a chronic cranial window. $PO_2$ imaging was performed within a 500 × 500 μm² field of view (FOV) down to 600 μm below the cortical surface. At each imaging depth, we selected the points for measuring $PO_2$ inside the microvascular segments (one point per segment), including arterioles, venules and capillaries (*Figure 1f*). Point-based acquisition of $PO_2$ was conducted plane-by-plane from the cortical surface down to the cortical depth of 600 μm with inter-plane separation of 50 μm. The intravascular Mean-$PO_2$ was measured in all diving arterioles, venules and in the majority of branching arterioles, venules and capillaries (6544 vascular segments across n = 15 mice) within the FOV. In addition, capillary RBC flux, speed and line-density, as well as RBC-$PO_2$, InterRBC-$PO_2$ and EAT (please see the Materials and methods section for the details) were calculated in a large subset of capillary segments (978 capillaries across n = 15 mice). In each mouse, the measurements were grouped by the cortical layer and then averaged. Subsequently, the average measurements for each cortical layer were averaged over animals (please see the Materials and methods section for the details). Histograms of capillary Mean-$PO_2$, RBC flux, line-density and speed are presented in *Figure 1—figure supplement 1*. The measurement information (e.g. imaging depth, animal number and sample size) for the main analysis in *Figures 2–7* is provided in the corresponding figure legends, as well as summarized as a table included in *Supplementary file 1*. Please note that the RBC speed estimation in this work was model-based by assuming a constant RBC size

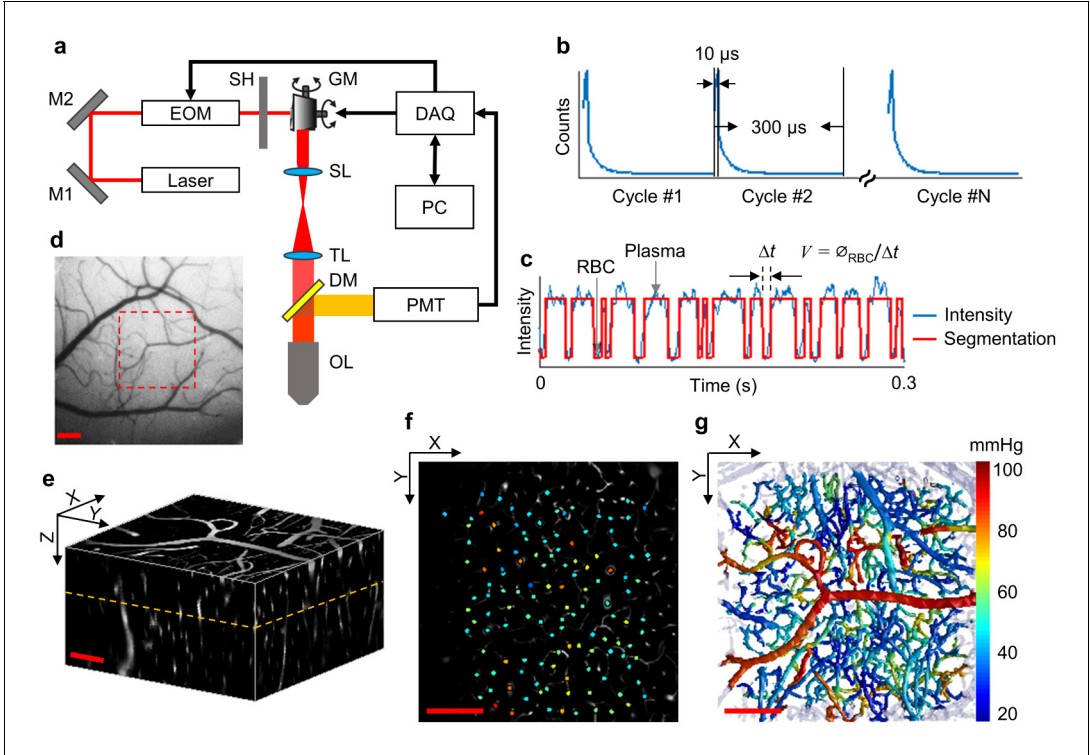

**Figure 1.** Experimental setup and data acquisition protocol. (**a**) Schematic of our home-built two-photon microscope. The components are abbreviated as: mirror (M), electro-optic modulator (EOM), shutter (SH), galvo mirrors (GM), scan lens (SL), tube lens (TL), dichroic mirror (DM), objective lens (OL), and photomultiplier tube (PMT). (**b**) Illustrative example of the phosphorescence decays for $PO_2$ recording at a single location. Each 300-$\mu$s-long cycle includes a 10-$\mu$s-long EOM-gated excitation, followed by a 290-$\mu$s-long detection of phosphorescence decay. (**c**) A representative phosphorescence intensity time course during $PO_2$ recording at a single location (blue curve). Each point in the time course represents the sum of the photon counts acquired during one 300-$\mu$s-long excitation/decay cycle in **b**. The red curve represents the binary segmented time course, with valleys and peaks representing RBC and blood-plasma-passages through the focal volume, respectively. (**d**) An image of the brain surface vasculature, taken through the chronic cranial window using a CCD camera. (**e**) The 3D representation of a Sulforhodamine-B-labeled cortical microvasculature imaged over the region of interest outlined by the red dashed square in d. f. $PO_2$ measurements inside the microvascular segments at the imaging plane outlined by the orange dashed line in **e**. $PO_2$ values (in mmHg, color-coded) were spatially co-registered with the microvascular angiogram. (**g**) Composite image shows the top view of the 3D projection of the $PO_2$ distribution in the microvascular network. Please note that panel **g** does not represent an instantaneous $PO_2$ distribution in the presented microvascular network. The color bar serves for panels **f** and **g**. Scale bars: 200 $\mu$m.

DOI: https://doi.org/10.7554/eLife.42299.002

The following source data and figure supplements are available for figure 1:

**Source data 1.** Measurements of Mean-$PO_2$ acquired in 6544 microvascular segments over n = 15 mice.
DOI: https://doi.org/10.7554/eLife.42299.005

**Figure supplement 1.** Histograms of capillary Mean-$PO_2$, RBC flux, line-density and speed.
DOI: https://doi.org/10.7554/eLife.42299.003

**Figure supplement 2.** Comparison between the capillary RBC speed measurements by the line-scan and point-scan methods.
DOI: https://doi.org/10.7554/eLife.42299.004

(6 $\mu$m) (*Unekawa et al., 2010*). However, the RBC size may vary with RBC speed, line-density and capillary diameter (*Chaigneau et al., 2003*). The comparisons between the RBC speed measurements obtained by using the model-based point-scan method and more direct measurements by the line-scan method (*Kleinfeld et al., 1998*) are presented in *Figure 1—figure supplement 2*.

The average $PO_2$ values in the diving arterioles and surfacing venules across cortical layers I-V are shown in *Figure 2a*. The $PO_2$ in the diving arterioles decreased from 99 $\pm$ 4 mmHg in layer I to 84 $\pm$ 3 mmHg in layer V; while the $PO_2$ in the surfacing venules exhibited a small increase starting from 43 $\pm$ 3 mmHg in layer V to 49 $\pm$ 4 mmHg in layer I. Similar trends were observed in the $SO_2$ values (*Figure 2b*), but the levels of $SO_2$ changes were different from those of $PO_2$ due to the sigmoidal shape of the oxygen-hemoglobin dissociation curve (*Uchida et al., 1998*). The $SO_2$ in the diving

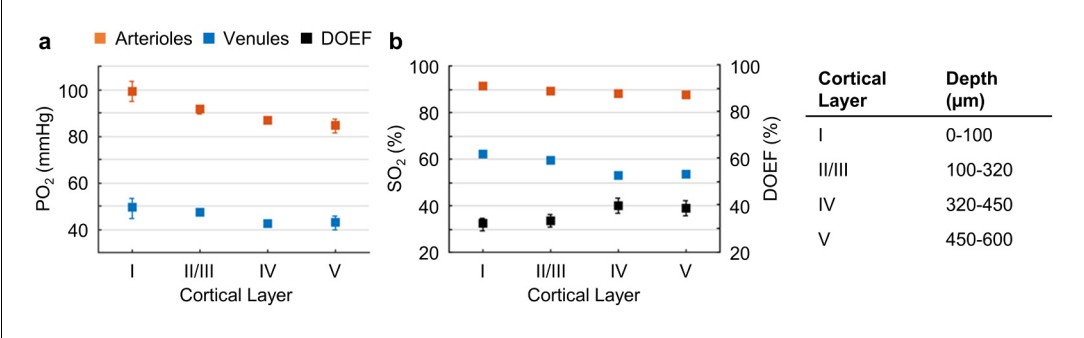

**Figure 2.** Cortical layer-dependent distributions of the arteriolar and venular intravascular $PO_2$ and $SO_2$. (a) Intravascular $PO_2$ in the diving arterioles (red symbols) and venules (blue symbols) across cortical layers I-V (11 arterioles, 14 venules, from n = 7 mice). (b) $SO_2$ in the diving arterioles (red symbols) and venules (blue symbols) across cortical layers I-V, and the depth-dependent OEF (DOEF, black symbols). For each diving arteriole or surfacing venule in a and b, $PO_2$ was tracked from the cortical surface down to the cortical depth of 600 μm. Data are expressed as mean ± SEM. Please note that the error bars may be too small to be visible.

DOI: https://doi.org/10.7554/eLife.42299.006

arterioles decreased slightly from 90.9 ± 0.8% in layer I to 87.1 ± 1.0% in layer V ($\Delta SO_{2,A}$ = 3.8 ± 1.2 %), while the $SO_2$ change in the surfacing venules was larger, from 53.5 ± 1.0% in layer V to 61.8 ± 0.8% in layer I ($\Delta SO_{2,V}$ = 8.3 ± 1.1 %). Consequently, the layer-specific difference between $SO_2$ in the diving arterioles and that in the surfacing venules increased toward deeper layers, and the depth-dependent OEFs (DOEF) in layers IV (39.9 ± 3.2%) and V (38.7 ± 3.3%) were larger than in more superficial layers, for example, layers I (32.0 ± 2.7%) and II/III (33.3 ± 2.7%; *Figure 2b*).

The observed increase in the DOEF with cortical depth suggests that surfacing venules received more oxygenated blood in the upper cortical layers than in the deeper layers, and that $SO_2$ in the pre-venular capillaries (or 'downstream capillaries') was higher for the capillaries joining venules in the upper cortical layers than for those in the deeper layers. We revisit this observation in the later section. In addition, the relation between blood flow, oxygen extraction, and the observed $SO_2$ changes across cortical layers was discussed.

## Capillary RBC flux and $PO_2$ are more homogenous in the deeper cortical layers

To better understand how the distributions of the capillary RBC flow and oxygenation change in order to support the heterogeneous demand for oxygen across cortical layers, we assessed both the spatial distributions and the temporal fluctuations of the resting capillary RBC flux and Mean-$PO_2$ in layers I-V. The spatial distributions for both the mean value and heterogeneity (quantified by STD and CV across capillaries) of capillary RBC flux and Mean-$PO_2$ in cortical layers I-V are shown in *Figure 3*. The RBC flux in layers IV (36 ± 4 RBC/s) and V (38 ± 6 RBC/s) were slightly lower than in layers I (41 ± 2 RBC/s) and II/III (41 ± 1 RBC/s; *Figure 3a*). The decrease in the RBC flux in the deeper layers might be due to the redistribution of RBCs over a denser capillary network, especially in layer IV (*Blinder et al., 2013*; *Sakadžić et al., 2014*), causing the RBC flux to be lower in individual capillaries. Importantly, both the STD and CV of RBC flux were lower in layers IV-V than in layers I-III, reaching a minimum in layer IV (*Figure 3b,c*); and the STD and CV of RBC flux in layer IV were significantly lower than their counterparts in layer I. This result suggests that RBC flux in the deeper cortical layers is more homogeneous, which may facilitate oxygen extraction as theoretically predicted (*Hartung et al., 2018*; *Jespersen and Østergaard, 2012*; *Schmid et al., 2017*). Similar trends could be observed in the distributions of capillary Mean-$PO_2$ (*Figure 3d–f*), capillary RBC-$PO_2$, InterRBC-$PO_2$, $SO_2$ and RBC speed (*Figure 3—figure supplement 1*). We did not find statistically significant difference in absolute RBC line-density, and its STD and CV between different cortical layers (*Figure 3—figure supplement 1*).

Next, we assessed the temporal fluctuations of RBC flux and Mean-$PO_2$ within individual capillaries, extracted from the 9-s-long acquisitions (*Figure 4*). The level of the temporal fluctuation (quantified by STD and CV) of capillary RBC flux (*Figure 4a,b*) decreased only in layer V, while the temporal

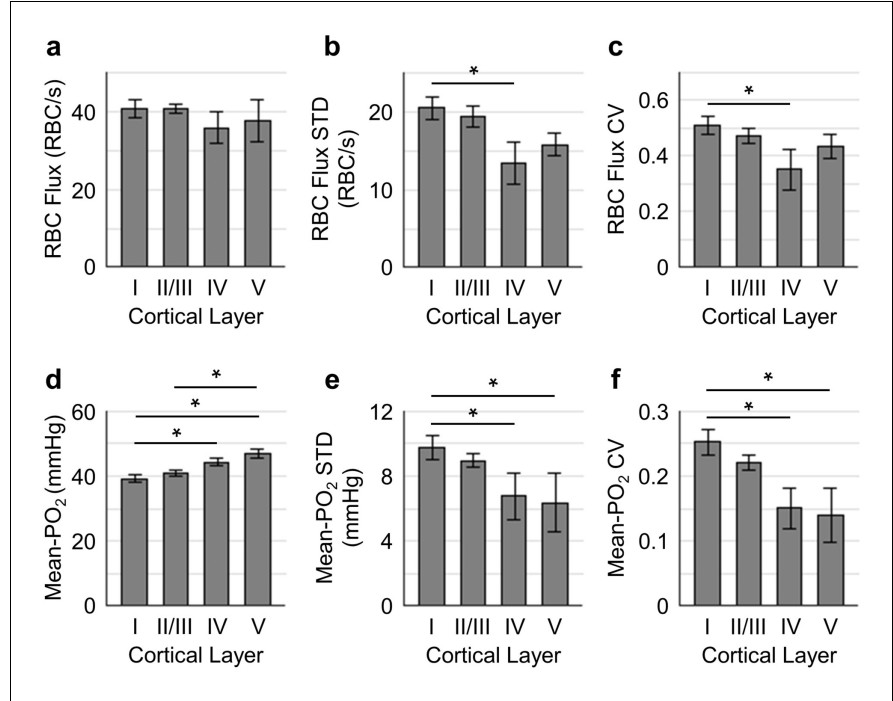

**Figure 3.** Spatial variations of capillary RBC flux and Mean-PO$_2$ as a function of cortical depth. The panels (**a–c**) and (**d-f**) show the dependence of the absolute values, standard deviations (STDs) and coefficients of variance (CVs) of capillary RBC flux and Mean-PO$_2$ on cortical layer, respectively. The absolute values, STDs and CVs were calculated across capillaries. The analysis in (**a–f**) was made with 400, 356, 118, and 104 capillaries measured in cortical layers I, II/III, IV and V, respectively, across n = 15 mice. Data are expressed as mean ± SEM. Statistical comparisons were carried out using ANOVA followed by Tukey HSD post hoc test. The asterisk symbol (*) indicates p<0.05.

DOI: https://doi.org/10.7554/eLife.42299.007

The following source data and figure supplement are available for figure 3:

**Source data 1.** Measurements of Mean-PO$_2$, RBC-PO$_2$, InterRBC-PO$_2$ and RBC flux acquired in 978 microvascular segments over n = 15 mice.
DOI: https://doi.org/10.7554/eLife.42299.009
**Figure supplement 1.** Distributions of capillary RBC-PO$_2$, InterRBC-PO$_2$, SO$_2$, RBC speed, and line-density, and their STDs and CVs as a function of cortical layer.
DOI: https://doi.org/10.7554/eLife.42299.008

fluctuation of the capillary Mean-PO$_2$ was significantly attenuated in layers IV and V (*Figure 4c,d*). Similar trend was also observed in RBC speed (*Figure 4—figure supplement 1*), but not in RBC line-density (not shown). However, the mean STDs and CVs of each observable were much smaller than that calculated across different capillaries (*Figure 3*).

Importantly, our measurements revealed that the Mean-PO$_2$, measured within individual capillaries as a function of time, was best correlated with RBC flux (*Figure 5*). We calculated the Pearson correlations between the temporal fluctuations (9-s-long traces with 0.6 s steps) of RBC flux, speed, line-density and the temporal fluctuation of the Mean-PO$_2$. All these parameters were simultaneously measured in each assessed capillary (n = 373 capillaries). The correlation coefficient *r* between the temporal fluctuations of the RBC flux and Mean-PO$_2$ (median value = 0.71) was higher than between the RBC speed and Mean-PO$_2$ (median value = 0.37), and between line-density and Mean-PO$_2$ (median value = 0.29). The *r* values were converted to Fisher *z* values to compare for statistical difference (*Diamond et al., 2006*). Mean-PO$_2$ was significantly stronger correlated with RBC flux than with RBC speed and line-density (*Figure 5c*). The results of the pairwise correlations between the temporal fluctuations of capillary RBC flux, speed and line-density are presented in *Figure 5—figure supplement 1*.

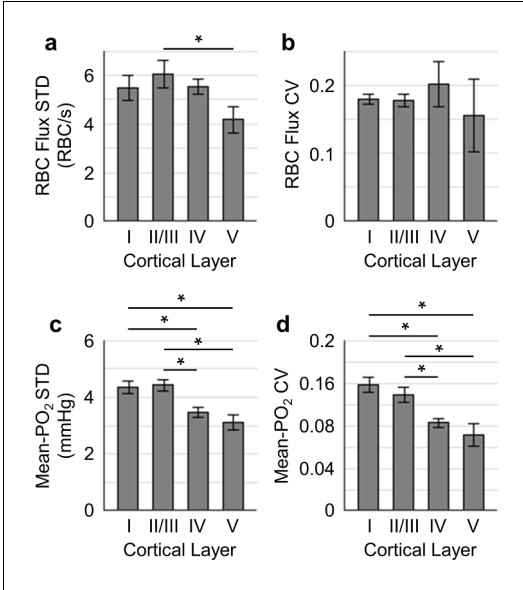

**Figure 4.** Temporal fluctuations of RBC flux and Mean-PO$_2$ within individual capillaries in cortical layers I-V. Panels (a–b) and (c–d) show the layer-dependent standard deviations (STDs) and coefficients of variance (CVs) of the temporal fluctuations of RBC flux and Mean-PO$_2$, respectively. The STD and CV of each observable for each capillary were calculated based on the 9-s-long time course. The analysis in (a–d) was made with 130, 140, 63 and 40 samples, collected in cortical layers I, II/III, IV and V, respectively, across n = 7 mice. Each sample corresponds to a 9-s-long, 15-time-point measurement acquired in each capillary. Data are expressed as mean ± SEM. Statistical comparisons were carried out using ANOVA followed by Tukey HSD post hoc test. The asterisk symbol (*) indicates p<0.05.
DOI: https://doi.org/10.7554/eLife.42299.010

The following figure supplement is available for figure 4:

**Figure supplement 1.** Temporal fluctuations of RBC speed within individual capillaries in cortical layers I-V.
DOI: https://doi.org/10.7554/eLife.42299.011

Our depth-resolved measurements across capillaries revealed that the distributions of RBC flux and Mean-PO$_2$ became more homogeneous in layers IV and V compared to layers I-III. This increase in the spatial homogeneity of the capillary RBC flow and oxygenation (*Figure 3*) was accompanied by an increase in the depth-dependent OEF (*Figure 2*). We also observed attenuation of the temporal fluctuations of these parameters in layers IV and V compared to layers I-III (*Figure 4*), although with much lower amplitudes than seen in the corresponding spatial variations (*Figure 3*). Finally, the time-course cross-correlation analysis revealed that the Mean-PO$_2$ was best correlated with the RBC flux, as opposed to RBC speed and line-density (*Figure 5*).

## Intracapillary resistance of oxygen transport to tissue decreases in deeper cortical layers

We now turn to the analysis of EAT that reflect the PO$_2$ modulation in capillaries due to the passages of individual RBCs. EAT has been observed in the peripheral and brain capillaries (*Barker et al., 2007*; *Golub and Pittman, 2005*; *Lecoq et al., 2009*; *Lecoq et al., 2011*; *Lyons et al., 2016*; *Parpaleix et al., 2013*). According to the modeling studies (*Barker et al., 2007*; *Golub and Pittman, 2005*; *Hellums, 1977*), larger EAT was associated with higher intracapillary resistance to oxygen transport to tissue from capillaries. Since cortical layers exhibit differences in the neuronal and vascular densities and possibly in oxygen metabolism, we examined whether EAT would differ in different cortical layers. Benefitting from the superior properties of Oxyphor2P, we were able to measure PO$_2$ in the blood plasma as a function of time (*Figure 6a*) and distance (*Figure 6—figure supplement 1*) from the center of the nearest RBC in a larger number of capillaries. In this work, PO$_2$ gradients were measured in a larger number of capillaries

(*Figure 6a*; 373 capillaries across n = 7 mice) and with greater imaging depth (from cortical surface to 600 µm) in comparison to the previous studies (*Lecoq et al., 2011*; *Lyons et al., 2016*; *Parpaleix et al., 2013*). Averaging over all the assessed capillaries, PO$_2$ (black curve in *Figure 6a*) decreased from 56.8 ± 0.4 mmHg at t = 0 ms to 28.6 ± 2.6 mmHg at t = 50 ms. The mean half-time-gap between adjacent RBCs was estimated to be 13.5 ms (denoted by the gray arrow in *Figure 6a*). The median values of capillary Mean-PO$_2$, RBC-PO$_2$, InterRBC-PO$_2$ and EAT were 39.4 mmHg, 49.2 mmHg, 36.3 mmHg and 11.7 mmHg, respectively (*Figure 6b*). Interestingly, we found a significant reduction in EAT in cortical layers IV (9.9 ± 0.8 mmHg) and V (11.0 ± 0.8 mmHg) compared to layers I (13.4 ± 0.6 mmHg) and II/III (13.3 ± 0.3 mmHg; *Figure 6c*), suggesting that the intracapillary resistance to oxygen transport to tissue may be lower in the deeper cortical layers, thus facilitating oxygen delivery to brain tissue according to the biophysical modeling (*Hellums, 1977*). In addition, we observed reduction in EAT STD and CV from layer I to layer V (*Figure 6—figure supplement 1*).

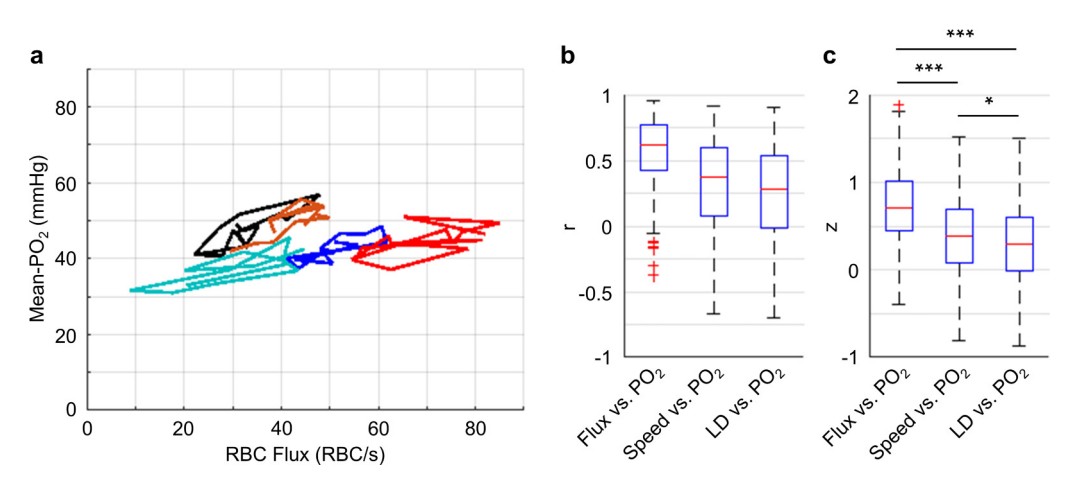

**Figure 5.** Correlations between the temporal fluctuations of capillary RBC flux, speed, line-density and Mean-PO$_2$. (a) Temporal evolutions (9-s-long traces with 0.6 s steps) of RBC flux and Mean-PO$_2$ from five representative capillaries. For each capillary, its Mean-PO$_2$ as a function of RBC flux is represented by a different line color; the consecutive time points are connected to illustrate the temporal trajectory of the variation. (b) Boxplots of the pairwise correlation coefficients (r) between the temporal fluctuations of capillary RBC flux, speed, line-density (LD) and Mean-PO$_2$. (c) Boxplots of the Fisher z values calculated based on the r values in **b**. The analysis in **b** and **c** was made with 373 capillaries, collected in cortical layers I-V, across n = 7 mice. Statistical comparisons were carried out using ANOVA followed by Tukey HSD post hoc test. The single-asterisk symbol (*) indicates p<0.05; the triple-asterisk symbol (***) indicates p<0.0001.

DOI: https://doi.org/10.7554/eLife.42299.012

The following figure supplement is available for figure 5:

**Figure supplement 1.** Quantifications of the temporal fluctuations of capillary RBC flux, speed, line-density and Mean-PO$_2$.

DOI: https://doi.org/10.7554/eLife.42299.013

## Low oxygen extraction along the superficial capillary paths contributes to the increase in the mean venular SO$_2$ toward cortical surface

To better understand why SO$_2$ in the ascending venules increased toward the cortical surface (*Figure 2*), we investigated the distributions of capillary flow and oxygenation along the capillary paths in the upper cortical layers. For this analysis, capillaries in the top 300 μm of the cortex (layers I-III) were grouped based on their branching orders into two main groups: 1) 'upstream' capillaries with branching orders A1-A3, which are closer to the arteriolar side of the network, and 2) 'downstream' capillaries with branching orders V1-V3, which are closer to the venular side of the network. Due to the limited number of capillary segments with assigned branch orders, A1-A3 and V1-V3 capillaries were grouped together to enable group comparisons with stronger statistical power.

The average values of Mean-PO$_2$, SO$_2$, EAT, RBC flux and line-density from the combined A1-A3 (upstream) and V1-V3 (downstream) capillaries are presented in *Figure 7a–e*. The average Mean-PO$_2$ in the A1-A3 capillaries (64.4 ± 2.4 mmHg) in layers I-III was, as expected, significantly higher than that in the V1-V3 capillaries (41.8 ± 0.9 mmHg; *Figure 7a*), suggesting that a large fraction of oxygen has been extracted along the capillary paths (i.e. from A1 to V1). Indeed, the average SO$_2$ in the A1-A3 capillaries (77.6 ± 1.7%) was significantly higher than that in the V1-V3 capillaries (67.3 ± 1.2%; *Figure 7b*). Provided that the average SO$_2$ in the A1 capillary segments was 81.0 ± 2.3%, and that the SO$_2$'s in the arterioles (SO$_{2,A}$) and venules (SO$_{2,V}$) at cortical surface were 91.0 ± 0.8% and 62.0 ± 0.8%, respectively (*Figure 2b*), we estimated that the decrease in SO$_2$ from the pial arterioles to the A1 capillary segments in the upper 300 μm of the cortex accounted for 34% (or 1/3) of the total extracted oxygen. Here, the total extracted oxygen was calculated as the A-V difference in SO$_2$ at the cortical surface (ΔSO$_{2,A-V}$ = 30 %; *Figure 2b*). Therefore, 66% (or 2/3) of the oxygen extraction in awake mice took place after the arterioles. This is in contrast to our previous study in anesthetized mice, which reported that ~ 50% of the oxygen delivered to brain tissue was extracted after the arterioles (*Sakadžić et al., 2014*). The average V1 capillary SO$_2$ in layers I-III was 65 ± 1.9%, higher than that in the ascending venules, both in layer I (62 ± 0.8%) and layer II/III (59 ± 0.4%; *Figure 2b*).

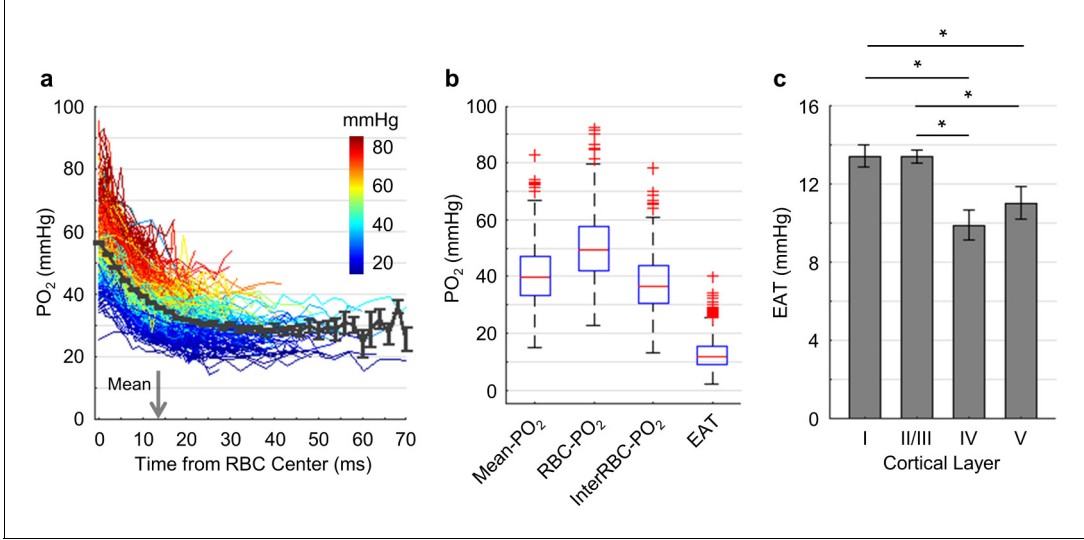

**Figure 6.** Dependence of EAT on cortical layer. (**a**) Intracapillary $PO_2$ gradients. The $PO_2$ gradients measured in different capillaries (373 capillaries across n = 7 mice) are color-coded based on their Mean-$PO_2$ values. The black curve represents the average $PO_2$ gradient. The gray arrow denotes the mean half-time-gap between adjacent RBCs (13.5 ms). (**b**) Boxplots of capillary Mean-$PO_2$, RBC-$PO_2$, InterRBC-$PO_2$ and EAT. (**c**) Dependence of EAT on cortical layer. The analysis in **c** was made with 130, 140, 63 and 40 capillaries measured in cortical layers I, II/III, IV and V, respectively, across n = 7 mice. Data are expressed as mean ± SEM. Statistical comparisons were carried out using ANOVA followed by Tukey HSD post hoc test. The single-asterisk symbol (*) indicates p<0.05.

DOI: https://doi.org/10.7554/eLife.42299.014

The following figure supplement is available for figure 6:

**Figure supplement 1.** Dependence of EAT STD and CV on cortical layer.

DOI: https://doi.org/10.7554/eLife.42299.015

Accordingly, capillaries that feed the surfacing venules in layers I-III apparently do so with more oxygenated blood, contributing to the increase in the venular $SO_2$ toward brain surface (**Figure 2b**).

The RBC flux in the A1-A3 capillaries (97 ± 6 RBC/s) was ~2.7 times higher than that in the V1-V3 capillaries (36 ± 2 RBC/s; **Figure 7d**). Since blood flow obeys the principle of mass conservation (i.e. the numbers of RBCs entering the capillary paths per unit time on the arteriolar side and exiting from the venous side must be equal), this result is in agreement with the previously observed ~3 fold-greater number of V1 than A1 capillaries in mouse cortex (**Nguyen et al., 2011**). Furthermore, the RBC speed in the A1-A3 capillaries (1.9 ± 0.2 mm/s) had a similar ratio (~3.2 times) to that in the V1-V3 capillaries (0.6 ± 0.1 mm/s; **Figure 7—figure supplement 1**), which is also consistent with the strong correlation between RBC flux and speed (**Figure 7—figure supplement 2**) (**Desjardins et al., 2014**; **Kleinfeld et al., 1998**).

We found strong positive correlations between the capillary RBC flux and both Mean-$PO_2$ and $SO_2$ in the downstream (V1-V3) capillaries, but not in the upstream (A1-A3) capillaries (**Figure 7f,g**), suggesting that a positive correlation between the RBC flux and oxygenation may be gradually building up along the capillary paths. In addition, we observed very heterogeneous distributions of both Mean-$PO_2$ (from ~11 mmHg to ~68 mmHg) and RBC flux (from ~30 RBCs/s to ~110 RBCs/s) in the V1 capillaries (**Figure 7h,i**). Lastly, the V1 capillary RBC flux was correlated positively with the ratio of the V1 capillary Mean-$PO_2$ to the $PO_2$ in the adjacent post-capillary venules (PCV $PO_2$) (**Figure 7j**). Therefore, heterogeneous oxygen delivery was taking place along different capillary paths, such that V1 capillaries had various levels of RBC flux and $PO_2$. However, the V1 capillaries with higher RBC flux were better oxygenated, contributing more to the increase in the PCV oxygenation.

Finally, EAT observed in the A1-A3 capillaries (10.9 ± 1.9 mmHg) was noticeably smaller than in the V1-V3 capillaries (16.6 ± 2.3 mmHg; **Figure 7c**), although this difference did not reach statistical significance (p=0.09). From both this work (**Figure 7—figure supplement 3**) and a previous study (**Lyons et al., 2016**), EAT did not exhibit any obvious dependence on the Mean-$PO_2$, RBC flux or

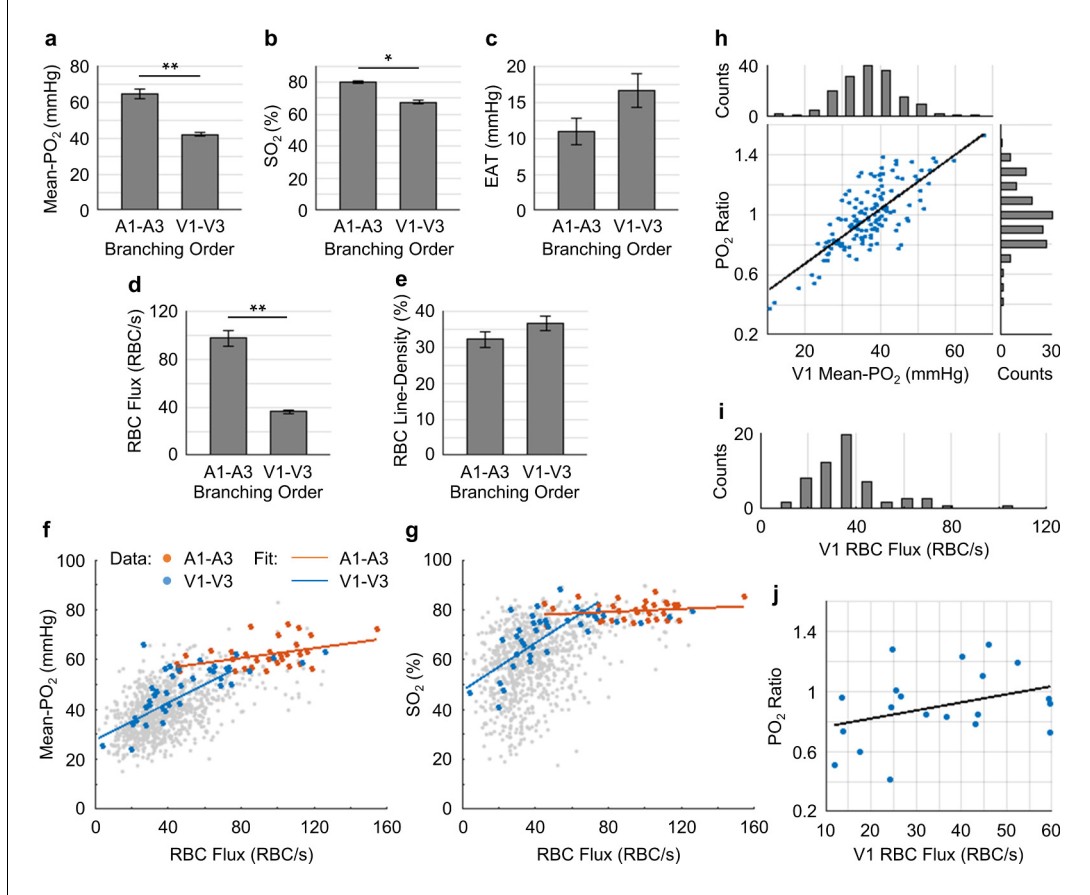

**Figure 7.** Capillary flow and oxygenation in the upstream and downstream branches. (a-e) Average capillary Mean-$PO_2$, $SO_2$, EAT, RBC flux and line-density, in the upstream (**A1–A3**) and downstream (**V1–V3**) capillary branches, across cortical layers I-III. Data are expressed as mean ± SEM. Statistical comparisons were carried out using Student's t-test. The single-asterisk symbol (*) indicates p<0.05; the double-asterisk symbol (**) indicates p<0.001. (**f** and **g**) Correlations between capillary RBC flux and Mean-$PO_2$ (**f**) and $SO_2$ (**g**). Data points and regression lines from the A1-A3, V1-V3, and all capillary segments (branching order unassigned) are color-coded red, blue, and gray, respectively. Linear regression slopes in **f**: V1-V3 slope = 0.37 mmHg•s•$RBC^{-1}$ ($R^2$ = 0.61), A1-A3 slope = 0.1 mmHg•s•$RBC^{-1}$ ($R^2 ≈ 0.17$). Linear regression slopes in **g**: V1-V3 slope = 0.50 s•$RBC^{-1}$ ($R^2$ = 0.52), A1-A3 slope = 0.03 s•$RBC^{-1}$ ($R^2 ≈ 0.04$). The analysis in **a–g** was made with 47 upstream and 50 downstream capillaries, across n = 5 mice. (**h**) Correlation between the $PO_2$ ratio (the V1 capillary Mean-$PO_2$ to the adjacent PCV $PO_2$) and the V1 capillary Mean-$PO_2$. Histograms of the V1 capillary Mean-$PO_2$, and $PO_2$ ratio are at the top and on the right from the main panel, respectively (178 capillaries, across n = 5 mice). (**i**) Histogram of the V1 capillary RBC flux (65 capillaries, across n = 5 mice). (**j**) Correlation between the V1 capillary RBC flux and $PO_2$ ratio (20 capillaries, across n = 5 mice). The linear regression slope = 0.01 s•$RBC^{-1}$ ($R^2 ≈ 0.12$).

DOI: https://doi.org/10.7554/eLife.42299.016

The following figure supplements are available for figure 7:

**Figure supplement 1.** Average capillary RBC speed in the upstream (**A1–A3**) and downstream (**V1–V3**) capillary branches, across cortical layers I-III.
DOI: https://doi.org/10.7554/eLife.42299.017

**Figure supplement 2.** Pairwise relations between capillary RBC flux, speed, line-density and Mean-$PO_2$.
DOI: https://doi.org/10.7554/eLife.42299.018

**Figure supplement 3.** Relations between capillary RBC line-density, Mean-$PO_2$, flux, speed and EAT.
DOI: https://doi.org/10.7554/eLife.42299.019

**Figure supplement 4.** Identification of a capillary segment having stalled RBC flow.
DOI: https://doi.org/10.7554/eLife.42299.020

**Figure supplement 5.** Identification of a suspected thoroughfare capillary.
DOI: https://doi.org/10.7554/eLife.42299.021

speed, but they were correlated negatively with line-density. Nevertheless, the observed trend that the average EAT in the A1-A3 capillaries was smaller than that in the V1-V3 capillaries is unlikely to be related to line-density, as the difference between the upstream and downstream line-density is insignificant (*Figure 7e*).

## Discussion

We have performed measurements of absolute intra-vascular $PO_2$ in arterioles, venules and a large number of capillaries (6544 capillaries across n = 15 mice), as well as EAT, RBC flux, speed and line-density in a subset of capillaries (978 capillaries across n = 15 mice). These parameters were measured simultaneously in the whisker barrel cortex in head-restrained awake C57BL/6 mice and thus were free of the confounding effects of anesthesia on neuronal activity, CBF and brain metabolism.

A new two-photon-excitable phosphorescence oxygen probe – Oxyphor2P was used in this study. Oxyphor2P belongs to the class of dendritically protected oxygen probes (*Lebedev et al., 2009*), and it is built around a newly developed Pt tetraarylphthalimidoporphyrin (PtTAPIP) (*Esipova et al., 2017*). In comparison to its predecessor (PtP-C343) (*Finikova et al., 2008*), Oxyphor2P exhibits red-shifted two-photon excitation and emission maxima, higher phosphorescence quantum yield and larger two-photon absorption cross-section (*Esipova et al., 2019*). Furthermore, in contrast to PtP-C343 the phosphorescence decay of Oxyphor2P exhibits well-defined single-exponential kinetics, which helps improve the accuracy of $PO_2$ calculation. These improvements of Oxyphor2P facilitated large-scale sampling of both capillary RBC flow and $PO_2$ deeper in the brain, for example, 600 μm vs. 450 μm (*Lyons et al., 2016*; *Sakadžić et al., 2014*), and with higher signal-to-noise ratios, which, in turn, powered the statistical analyses beyond what could be achievable with PtP-C343 (*Devor et al., 2011*; *Lecoq et al., 2011*; *Lyons et al., 2016*; *Parpaleix et al., 2013*; *Sakadzić et al., 2010*; *Sakadžić et al., 2014*). In addition, the very high phosphorescence quantum yield of Oxyphor2P (*Esipova et al., 2017*; *Esipova et al., 2019*) enabled detection of RBC-passages in capillaries based on phosphorescence alone, without the need to enhance the signal from the blood plasma using additional chromophores (*Lyons et al., 2016*).

We first assessed oxygenation in the diving arterioles and surfacing venules and observed a decrease in $PO_2$ with cortical depth for both vessel types (*Figure 2a*). The $PO_2$ decrease in the diving arterioles between layer I and V was about 15 mmHg, equivalent to the difference in $SO_2$ of 3.8%. This modest oxygen extraction along the diving arterioles implies that much more oxygen was extracted from the rest of the microvascular network, including arteriolar branches, capillaries and potentially venules. Previous measurements performed in mice (*Kisler et al., 2017*; *Moeini et al., 2018*) and rats (*Devor et al., 2011*; *Sakadžić et al., 2016*) reported pronounced $PO_2$ gradients in the periarteriolar tissue around the cortical diving arterioles, implying extraction of oxygen from cortical diving arterioles. Specifically, a significantly steeper decrease in $PO_2$ with cortical depth was observed directly in diving arterioles in isoflurane-anesthetized mice by *Kazmi et al. (2013)* and indirectly, based on the depth dependence of the extravascular (tissue) $PO_2$ measured immediately next to the diving arterioles, in α-chloralose anesthetized rats by *Devor et al. (2011)*. The most likely reason for these discrepancies is the higher suppression of CBF than $CMRO_2$ under different anesthesia regimes. In awake mice, *Lyons et al. (2016)* measured approximately constant $PO_2$ in the diving arterioles over cortical depth of 0–400 μm. This result is consistent with our current observation that in awake mice, the amount of oxygen extracted along the diving arterioles over depth represents a small fraction of the total amount of oxygen transported by these arterioles.

In contrast to the moderate $PO_2$ decrease along the penetrating arterioles, the decrease in the venular $PO_2$ from layer I to V was much smaller (~6 mmHg), but resulted in a larger fractional decrease in $SO_2$ (8.3%) than in the penetrating arterioles (*Figure 2b*). An increase in $PO_2$ in ascending venues towards the cortical surface has been previously observed in isoflurane-anesthetized mice (*Sakadžić et al., 2010*). However, the $PO_2$ values, measured previously along the ascending venules in the fore-paw and hind-paw regions of the somatosensory cortex in awake mice, were reported to be nearly constant (*Lyons et al., 2016*). The difference between these observations may be attributed to the difference in the studied cortical regions and/or to the lower measurement accuracy associated with the old probe – PtP-C343, because of its lower brightness and inability to resolve small $PO_2$ changes. In addition, some discrepancies in the absolute baseline $PO_2$ in arterioles and venules at different depths were observed between our present and some previous studies

(*Kazmi et al., 2013*; *Lecoq et al., 2011*; *Lyons et al., 2016*; *Parpaleix et al., 2013*; *Sakadžić et al., 2010*). Different anesthesia regimes, different cortical areas and, most importantly, use of anesthetized vs. awake animals would be the most obvious factors underlying the observed discrepancies. Furthermore, cortical temperature typically was not controlled in the previous studies, while the presence of a cranial window can cause reduction in the cortical temperature (*Shirey et al., 2015*), especially when imaging is carried out using a non-heated water-immersion objective. Temperature affects all physiological parameters and chemical properties, including $O_2$ diffusion and solubility, hemoglobin affinity to $O_2$, CBF, $CMRO_2$ and the triplet decay time of the oxygen probe (*Benesch et al., 1969*; *Croughwell et al., 1992*; *Finikova et al., 2008*; *Hanks and Wallace, 1949*; *Jones and Siegel, 1969*; *Pray, 1952*; *Rosomoff and Holaday, 1954*; *Rossing and Cain, 1966*; *Soukup et al., 2002*) – all of which may affect the experimental results.

A faster decrease in $SO_2$ with cortical depth in the ascending venules than in the penetrating arterioles resulted in a higher depth-dependent OEF in the deeper cortical layers, reaching the maximum in layer IV (*Figure 2b*). In addition, the total blood flow may be higher in layer IV than in layer I, as suggested by the measurements of capillary RBC flux (~13% lower in layer IV than in layer I; *Figure 3a*) and capillary segment density (~50% higher in layer IV than in layer I; *Gould et al., 2017*; *Sakadžić et al., 2014*). Altogether, this implies that oxygen extraction was higher in the deeper cortical layers, which would be in agreement with the findings that layer IV of mouse cortex has the highest neuronal and capillary densities (*Blinder et al., 2013*; *Lefort et al., 2009*; *Patel, 1983*; *Wu et al., 2016*), and that the cells in layer IV exhibit the highest cytochrome oxidase labeling activity, suggestive of the highest oxidative metabolism (*Land and Simons, 1985*). We estimated the depth-dependent OEF, which implies existence of a laminar flow pattern in the brain cortical microvascular network. This was recently confirmed by *Schmid et al. (2017)*, who applied numerical modeling of blood flow and tracking of the trajectories of individual RBCs in realistic mouse cortical vasculature, which led to conclusion that RBCs predominantly flow in plane and no significant RBC flow in the direction of cortical depth takes place. Since oxygen extraction depends on both total blood flow and arterio-venous oxygen saturation difference, it will be important in the future to further experimentally investigate the total blood flow differences across different cortical layers.

The mean capillary RBC flux decreased only slightly with cortical depth, without reaching statistically significant difference across layers I-V (*Figure 3a*). This might be due to the redistribution of the blood flow over the denser capillary network in the deeper cortical layers, especially in layer IV (*Blinder et al., 2013*; *Sakadžić et al., 2014*; *Wu et al., 2016*), causing the RBC flux to be lower in individual capillaries. The observed mean capillary RBC flux (41.5 ± 1.2 RBCs/s; *Figure 1—figure supplement 2*) is in good agreement with the value of 41.9 ± 1.8 RBCs/s, reported by *Lyons et al. (2016)*, and somewhat lower than ~48 RBC/s reported by *Moeini et al. (2018)* for somatosensory cortex in awake mice. The average capillary Mean-$PO_2$ increased slightly by ~7 mmHg from layer I to V (*Figure 3d*), which is in agreement with the previously reported increase in capillary Mean-$PO_2$ in both awake (*Lyons et al., 2016*) and isoflurane-anesthetized mice (*Sakadžić et al., 2014*) across the upper 400 µm of the cortex. One potential explanation for this increase could be related to the finding by *Blinder et al. (2013)*, who reported that, in the upper 600 µm of somatosensory cortex, the probability of penetrating arterioles giving off side branches increased with cortical depth, whereas the probability for the ascending venules decreased with depth. This suggests that the ratio between the number of the more oxygenated (upstream) capillary branches and that of the less oxygenated (downstream) capillary branches increases with depth, contributing to the higher capillary $PO_2$ at greater depth.

Importantly, we found that the distributions of capillary RBC flux, speed and Mean-$PO_2$ were more homogeneous in cortical layers IV and V (*Figure 3*). The layer-dependent homogeneity of blood flow has been predicted by modeling the blood flow distribution in large-scale mouse brain microvasculature covering the whisker barrel cortex (*Hartung et al., 2018*). Both analytical and numerical models of oxygen advection and diffusion in microvascular networks predict that more homogenous capillary blood flow facilitates oxygen delivery to tissue (*Jespersen and Østergaard, 2012*; *Østergaard et al., 2013*; *Østergaard et al., 2014a*; *Østergaard et al., 2014b*; *Rasmussen et al., 2015*). Indeed, capillary blood flow homogeneity, measured by the distribution of capillary transit time or RBC flux during functional activation in the brain in healthy anesthetized animals, have been reported previously (*Gutiérrez-Jiménez et al., 2016*; *Lee et al., 2016*; *Stefanovic et al., 2008*). Interestingly, capillary flow homogeneity was absent in the mouse model

of Alzheimer's disease (*Gutiérrez-Jiménez et al., 2018*), while increased capillary blood flow heterogeneity was found in patients of Alzheimer's disease (*Eskildsen et al., 2017*; *Nielsen et al., 2017*), suggesting a possible link between disturbed capillary flow patterns, reduced oxygen supply to tissue, and the progression of neurodegeneration. Our observations are in agreement with the previously reported lower capillary transit time heterogeneity in cortical layers IV and V than in the upper layers in anesthetized mice (*Merkle and Srinivasan, 2016*). Furthermore, capillary Mean-PO$_2$ also exhibited a similar homogeneous trend (*Figure 3e,f*). This could be expected since capillary RBC flow is correlated positively with Mean-PO$_2$ (*Figure 7—figure supplement 2*) (*Lyons et al., 2016*).

The temporal resolution of our measurements (0.6 s) ensured that the blood flow and PO$_2$ fluctuations due to cardiac and respiratory cycles, as they occurred to awake mice, were averaged out, but it is sufficient to capture the dynamics related to the rate of oxygen consumption, which could be estimated as the time for tissue PO$_2$ to drop to zero after blood flow stoppage. As reported in cats, that time was at least several seconds (*Acker and Lübbers, 1977*; *Whalen and Nair, 1975*). In addition, blood flow responses to neuronal activation could typically be resolved with such temporal resolution (*Uhlirova et al., 2016*). However, it is important to note that our experiments probed just one frequency window within a wide range of both faster and slower fluctuations present in the microvascular network. The cause of the dampening of the temporal fluctuations with depth is unclear. It may be related to the anatomical and functional differences between cortical layers; it also can be due to the expected dampening along the vascular paths if the temporal fluctuations originate from the upstream arteriolar blood flow.

We observed strong positive correlations between the temporal fluctuations of capillary RBC flux and Mean-PO$_2$ (*Figure 5*). This observation is in agreement with the previously reported positive correlation between the fluctuations of RBC flux and extravascular (tissue) PO$_2$ measured adjacent to the capillaries in rat tumor, although the latter occurred at a lower frequency range (*Braun et al., 1999*; *Kimura et al., 1996*). Importantly, the fluctuation of capillary Mean-PO$_2$ was found to be correlated positively with the fluctuation of RBC flux significantly stronger than with the fluctuations of RBC speed or line-density (*Figure 5b,c*), although these three variables are closely related as RBC flux is proportional to the product of RBC speed and line-density (*Kleinfeld et al., 1998*). This result could be expected, since most oxygen in the blood is bound to the hemoglobin inside RBCs (*Pittman, 2011*). We also observed strong positive correlation between the fluctuations of RBC flux and speed, but the fluctuations of line-density was poorly correlated with both of them (*Figure 5—figure supplement 1*). These correlations between the temporal fluctuations are in agreement with the trends measured using the populations of capillaries (*Figure 7—figure supplement 2*) as well as with the previous studies (*Kleinfeld et al., 1998*; *Santisakultarm et al., 2012*; *VanTeeffelen et al., 2008*). Despite poor correlation between the RBC flux and line-density, we observed reasonably positive correlation between the temporal fluctuations of capillary Mean-PO$_2$ and line-density (*Figure 5*). Similarly, shown in *Figure 7—figure supplement 2* and in *Lyons et al. (2016)*, capillary Mean-PO$_2$ and line-density, measured in the populations of capillaries, were also positively correlated, emphasizing the specific role of RBC line-density in oxygen transport (*Lücker et al., 2017*).

Capillary PO$_2$ gradients were measured to assess the intracapillary resistance to oxygen delivery to tissue and to calculate capillary SO$_2$ (*Figure 6*). For measuring EAT, we followed the previously developed methods (*Lecoq et al., 2011*; *Lyons et al., 2016*; *Parpaleix et al., 2013*). Compared to the previous studies performed with the old probe – PtP-C343 (*Lecoq et al., 2011*; *Lyons et al., 2016*; *Parpaleix et al., 2013*), the superior properties of Oxyphor2P enabled us to measure PO$_2$ gradients in a much larger number of capillaries over a greater cortical depth (down to 600 μm below the cortical surface) and using a shorter acquisition time (9 s) per measurement location. In agreement with *Lyons et al. (2016)*, we observed EAT in capillaries (*Figure 6*), although the magnitude was lower than previously reported. EAT did not appear to depend significantly on RBC flux, speed or Mean-PO$_2$, but were correlated negatively with the line-density (*Figure 7—figure supplement 3*). Our measurements are in agreement with the previous observations (*Lyons et al., 2016*) as well as with the theoretical predictions (*Hellums, 1977*; *Lücker et al., 2017*). Importantly, we detected a significant decrease in EAT towards the deeper cortical layers (*Figure 6c*). Since EAT is directly related to the intravascular resistance to the diffusive oxygen transport from RBCs to tissue (*Hellums, 1977*), reduced EAT in the deeper layers represents another example of the adaptation of the microvascular network in order to facilitate local oxygen delivery in cortical regions with higher oxygen demand (*Pries et al., 1994*; *Pries et al., 1998*; *Reglin et al., 2009*; *Reglin et al., 2017*).

However, it is important to note that mechanisms that govern EAT are multifactorial. EAT may be affected by multiple parameters, such as RBC spacing, shape and wall-to-RBC spatial clearance (*Golub and Pittman, 2005*; *Lücker et al., 2017*; *Popel, 1989*). In addition, our EAT measurements provide average values based on multiple RBC-passages. Therefore, they do not account for fluctuations of the parameters that affect EAT, which may differ between cortical layers and/or proximal and distal capillaries, and differentially affect the EAT measurements. Smaller difference between RBC-$PO_2$ and InterRBC-$PO_2$ and, consequently, smaller EATs at greater depths are likely a consequence of more narrowly distributed capillary blood flow and oxygenation. However, it is less clear why the mean RBC-$PO_2$ and InterRBC-$PO_2$ are increasing with depth (*Figure 3—figure supplement 1*). The observed increases in the mean RBC-$PO_2$ and InterRBC-$PO_2$ may be in part due to the potential change in proportion of the more oxygenated upstream capillaries to the less oxygenated downstream capillaries. Our measurement protocol (i.e., measuring $PO_2$ in most of the capillaries identified within the field of view at each imaging plane) together with the large number of capillaries interrogated at each depth, ensured 1) that the upstream and downstream capillaries were sampled proportional to their number densities, and 2) that the mean $PO_2$ values were properly estimated, but could not guarantee an equal sampling of the upstream and downstream capillaries unless the equal distributions of the upstream and downstream capillaries are naturally occurring. Other mechanisms, such as the effect of the increased homogeneity of capillary RBC flux and $PO_2$ on the mean intravascular $PO_2$, may also contribute.

The elevated $SO_2$ in the surfacing venules in layers I-III suggests that in the upper cortical layers (e.g. layers I-III) the downstream capillaries and post-capillary venules that feed the surfacing venules are more oxygenated compared to the surfacing venules in the deeper layers (e.g. layers IV and V). We therefore assigned branching orders to the capillary segments in cortical layers I-III and investigated the distributions of capillary RBC flow and oxygenation along the capillary paths. We found that the average $SO_2$ in the V1 capillaries selected across layers I-III was indeed higher than the $SO_2$ in the ascending venules in both layers I and II/III, confirming that high oxygen content in the superficial capillaries contributed to the increase in the venular $SO_2$ toward brain surface. This result is in line with the recent experimental and theoretical findings that capillary transit time is shorter in the surface cortical layers (*Gutiérrez-Jiménez et al., 2016*; *Schmid et al., 2017*), causing less efficient release of oxygen by the RBCs, thus elevating the $SO_2$ in the surfacing venules (*Schmid et al., 2017*). This result also suggests that the baseline oxygen extraction may be different between the cortical layers, which has been predicted by several theoretical studies using realistic vascular anatomical models (*Gould and Linninger, 2015*; *Gould et al., 2017*; *Linninger et al., 2013*). If true, this may have implications on the interpretation of results from different imaging modalities such as BOLD fMRI (*Blockley et al., 2015*; *Griffeth and Buxton, 2011*; *Siero et al., 2011*; *Silva and Koretsky, 2002*; *Vazquez et al., 2006*; *Yu et al., 2014*). Secondly, approximately 1/3 of the total oxygen extraction took place along the paths from the pial arterioles to the A1 capillary segments in layers I-III. This observation differs from our previous study in isoflurane-anesthetized mice, where we found, based on the measurements in the upper 450 µm of somatosensory cortex, that ~50% of the oxygen delivered to brain tissue was extracted from arterioles (*Sakadžić et al., 2014*). The discrepancy might be due to the effect of anesthesia, which suppressed cerebral oxygen metabolism and possibly increased both CBF and arteriolar surface area due to vasodilation (*Alkire et al., 1999*; *Goldberg et al., 1966*; *Ogawa et al., 1990*). As a result of these perturbations, oxygen extraction could be shifted towards the upstream microvascular segments (*Sakadžić et al., 2014*). Based on the observed difference in oxygen extraction and the assumed difference in oxygen metabolism between cortical layers, we anticipate that in cortical layers IV and V, the fraction of the extracted oxygen from arterioles may be smaller than in layers I-III, but still significant.

Our data provided additional evidence that the distributions of capillary flow and oxygenation were highly heterogeneous and strongly positively correlated with one-another, in particular in the downstream capillaries (*Figure 7*). This implies that the mixed venous blood oxygenation was, therefore, a result of the contributions from the wide distribution of capillary paths, carrying blood that was both more and less oxygenated than the blood in the postcapillary venules (*Figure 7*). At the extreme ends of this distribution are capillary paths with very low and very high oxygenation and blood flow. The paths with very low oxygenation may be the first sites of coupling of microvascular dysfunction and progression of various brain pathologies (*Erdener et al., 2017*; *Lücker et al., 2018a*). Indeed, it has been recently reported that only ~0.4% of the cortical capillaries in healthy

mice had 'stalled flow', but in transgenic mice of Alzheimer's disease the fraction was as high as 2% (*Cruz Hernandez et al., 2016*; *Momjian-Mayor and Baron, 2005*). As an example of such poorly oxygenated/perfused capillaries, we identified a capillary segment with $PO_2$ of ~15 mmHg and no detectable RBC flow during the 9-s-long acquisition (*Figure 7—figure supplement 4*). In contrast, the paths with very high oxygenation and blood flow (*Figure 7—figure supplement 5*) may be especially good sites for implementing blood flow control in response to locally increased oxygen demand, since increasing their resistance to flow may quickly redistribute blood over the nearby capillary network. However, the prevalence of such highly oxygenated/perfused capillaries in the cortical microvascular network and the existence of mechanisms of their site-specific control are unknown (*Hudetz et al., 1996*). Further studies should address these questions in relation to both normal and pathological brain conditions.

One potential limitation of this study is that the assignment of the cortical layers as a function of the cortical depth was performed based on the literature data (*Blinder et al., 2013*; *Lefort et al., 2009*), as opposed to identification of the layer-specific anatomical landmarks. Nevertheless, while slight shifts in the layer boundaries, which may have resulted from such an assignment, may have affected the exact values, it is unlikely that the observed general trends and significant differences in the observables between the layers would have changed. In addition, layer V in mouse barrel cortex spans the depth range of 450–700 µm (*Blinder et al., 2013*; *Lefort et al., 2009*), so that the range of 450–600 µm, interrogated in this study, likely overlaps the best with the upper part of layer V (i.e., layer Va) (*Blinder et al., 2013*; *Lefort et al., 2009*).

Another limitation is that at greater cortical depths, expanding shadows below large pial vessels, partial obstruction of the optical paths by the edge of the cranial window, and gradual loss of resolution, contributed to the smaller sampling size. It is possible that due to the smaller number of measured capillaries in the deeper cortical layers (e.g., 104 capillaries in layer V vs. 400 capillaries in layer I; *Figure 3*), some investigated variables did not show statistically significant difference between cortical layers.

The capillaries were identified based on their morphology, without taking into account the smooth muscle cell coverage and pericyte types (*Attwell et al., 2016*; *Hall et al., 2014*; *Hartmann et al., 2015*; *Hill et al., 2015*; *Mishra et al., 2014*; *Peppiatt et al., 2006*; *Secomb, 2017*), and the assignment of capillary branching order indices was performed manually, which may be operator-dependent. In principle, misclassifying vessel types and/or branching orders could potentially influence our analysis. However, based on the overwhelmingly larger number of capillaries compared to the non-penetrating arterioles and venules, our conclusions in general are unlikely to be different. Furthermore, identification of capillaries based on the biochemical staining of smooth muscle cells and pericytes (*Hall et al., 2014*; *Hartmann et al., 2015*) will add significant complexity to the study, especially considering a large sample size used in our analysis, but in the end may not provide more accurate results, as the debate on what is a proper classification of vessel-types based on staining is still ongoing (*Hill et al., 2015*).

The RBC speed was calculated by assuming a constant RBC size (6 µm) (*Unekawa et al., 2010*) along the capillary axis, without considering its potential variations with RBC speed, line-density and capillary diameter. In a separate set of experiments, we performed line-scan measurements (*Kleinfeld et al., 1998*) in 58 capillaries (2-s-long acquisition, 2 kHz line-scan rate) in n = 2 awake mice, and obtained close mean RBC speed values by processing the data by using two methods: by the line-scan method described in *Kleinfeld et al. (1998)* (mean RBC speed = 0.61 mm/s) and by the RBC-passage (or point-scan) method used in this work (mean RBC speed = 0.69 mm/s). The mean RBC longitudinal size was estimated by following the procedures described in *Chaigneau et al. (2003)*, and it did not vary significantly as a function of RBC speed, line-density and capillary diameter, except for the fast RBCs (>1 mm/s), which were also associated with the noisier measurements by both techniques. However, the mean fluctuation of the RBC longitudinal size over the 2-s-long acquisition was moderate (STD = 2.3 ± 0.6 µm). Therefore, group comparison of the mean RBC speed values measured by the RBC-passage method may be conducted with reasonable accuracy, but the instantaneous speed of individual RBCs obtained by this method may have larger measurement errors (please see *Figure 1—figure supplement 2* and text for additional details). These limitations of the technique should be considered for particular experimental designs.

The RBCs that are touching each other might be counted mistakenly as a single RBC, causing underestimation of RBC flux and speed. We estimated that approximately 6% of the 'valleys' might

be caused by the passing of multiple RBCs through the optical focus. However, as no difference in RBC line-density across different cortical layers was observed, we do not anticipate that our layer-specific analysis is affected as it is unlikely that the RBCs-touching phenomenon differentially affects the measurements in different cortical layers.

We would also like to mention that some discrepancy exists between the EAT magnitudes measured using 2PLM by different groups. For example, rather large EAT was reported in *Lecoq et al. (2011)*; *Lyons et al. (2016)*; *Parpaleix et al. (2013)* with the PtP-C343 probe. Much smaller EAT (only a few mmHg) were measured previously using also PtP-C343 (unpublished data from *Sakadžić et al., 2014*), and moderate EAT are reported in the present work with Oxyphor2P. It should be mentioned that the data underlying the EAT measurements are typically noisier than those used to derive mean intravascular and tissue $PO_2$'s, since to quantify EAT the signals have to be split into multiple distance or time bins. In addition, the previously used probe PtP-C343 has intrinsically non-single-exponential phosphorescence decay and much lower emission quantum yield. These limitations, in combination with potentially slightly different implementations of acquisition protocols and algorithms for fitting the phosphorescence decay data, are likely to be the dominant factor contributing to the differences in the reported EAT magnitudes. By using Oxyphor2P, which has much stronger signal and much better defined single-exponential decay, making data analysis much more robust, here we greatly reduced uncertainty in the EAT measurements, which was inherent to the previous probe PtP-C343. Nevertheless, we still would like to emphasize the importance of aligning the acquisition and data analysis protocols across the labs as well as using the same data acquisition protocols during measurement as used for probe calibration.

We averaged all the phosphorescence decays within the valleys (i.e. RBC-passages) in the segmented phosphorescence intensity time courses to calculate RBC-$PO_2$. This approach does not consider that within the valley $PO_2$ may vary as a function of distance to the center of the valley, and the extent of this variation may be different among capillaries of different diameter and/or RBC speeds.

In conclusion, we have experimentally mapped the distributions of microvascular flow and oxygenation in the whisker barrel cortex in awake mice using two-photon phosphorescence lifetime microscopy. We have found evidence that oxygen was extracted differently in different cortical layers, and that the distributions of microvascular blood flow and oxygenation were adjusted across layers in a way that facilitated oxygen delivery in the deeper layers. Specifically, the depth-dependent OEF was measured higher in cortical layers IV and V, where the oxidative metabolism is presumably the highest in the cortex. This increase was accompanied by the more homogenous capillary RBC flow and oxygenation, as well as by the reduction of intracapillary resistance to oxygen diffusion to tissue (inferred from the EAT changes). In addition, we have found that arterioles in the superficial cortical layers (e.g. layers I-III) contributed significantly to the oxygen extraction from blood (34%) even in awake mice. We anticipate that our results and analysis will help better understand the normal brain physiology, and the progression of brain pathologies that affect cerebral microcirculation. They will also inform more accurate biophysical models of the cortical layer-specific oxygen delivery and consumption, as well as improve the interpretation of the results from other brain imaging modalities.

## Materials and methods

### Animal preparation

We used n = 15 C57BL/6 mice (3–5 months old, 20–25 g, female, Charles River Laboratories) in this study. We followed the procedures of chronic cranial window preparation outlined by *Goldey et al. (2014)*. A custom-made head-post (*Mateo et al., 2011*) allowing repeated head immobilization was glued to the skull, overlaying the right hemisphere. A craniotomy (round shape, 3 mm in diameter) was performed over the left hemisphere, centered approximately over the E1 whisker barrel. The dura was kept intact. The cranial window was subsequently sealed with a glass plug (*Komiyama et al., 2010*) and dental acrylic. After surgery, mice were given 5 days to recover before starting the habituation training. The training was conducted while mice were resting on a suspended soft fabric bed in a home-built platform, under the microscope. Mice were gradually habituated to longer periods (from 10 min to 2 hr) of head-restraint with the head slightly rotated (~35°) to make the cortical surface perpendicular to the optical axis. All mice were rewarded with sweetened

milk every 15 min during both training and experiments. While head-restrained, the mice were free to readjust their body position and from time to time displayed natural grooming behavior. All animal surgical and experimental procedures were conducted following the Guide for the Care and Use of Laboratory Animals and approved by the Massachusetts General Hospital Subcommittee on Research Animal Care (Protocol No.: 2007N000050).

## Two-photon microscope

In this study, we employed our previously developed home-built two-photon microscope (*Figure 1a*) (*Sakadzić et al., 2010*; *Yaseen et al., 2015*). Briefly, a pulsed laser (InSight DeepSee, Spectra-Physics, tuning range: 680 nm to 1300 nm,~120 fs pulse width, 80 MHz repetition rate) was used as an excitation source. Laser power was controlled by an electro-optic modulator (EOM). The laser beam was focused with a water-immersion objective lens (XLUMPLFLN20XW, Olympus, NA = 1.0), and scanned in the X-Y plane by a pair of galvanometer scanners. The objective was moved along the Z axis by a motorized stage (M-112.1DG, Physik Instrumente) for probing different cortical depth. Oxyphor2P was excited at 950 nm. The emitted phosphorescence, centered at 760 nm, was directed toward a photon-counting photomultiplier tube (H10770PA-50, Hamamatsu) by a dichroic mirror (FF875-Di01−25 × 36, Semrock), followed by an infrared blocker (FF01-890/SP-25, Semrock) and an emission filter (FF01-795/150-25, Semrock). The lateral and axial resolutions of the $PO_2$ measurements were estimated as 2 and 5 µm, respectively (*Sakadzić et al., 2010*; *Yaseen et al., 2015*). During the experiments, the objective lens was heated by an electric heater (TC-HLS-05, Bioscience Tools) to maintain the temperature of the water between the cranial window and the objective lens at 36–37°C. In addition, we attached the sensor of an accelerometer module to the suspended bed of the mouse platform to record the signals induced by mouse motion. Mice were continuously monitored during experiments by acquiring live videos with a CCD camera (CoolSNAPfx, Roper Scientific) using a LED illumination at 940 nm.

## Intravascular $PO_2$ imaging

Before imaging, mice were briefly anesthetized by isoflurane (1.5–2%, during ~2 min), and then the solution of Oxyphor2P (0.1 ml at ~34 µM) was retro-orbitally injected into the bloodstream. Here, we used a new phosphorescent oxygen probe – Oxyphor2P (*Esipova et al., 2019*). Compared to its predecessor – PtP-C343 (*Finikova et al., 2008*), Oxyphor2P exhibits red-shifted excitation and emission ($\lambda_{exc}$ = 950 nm, $\lambda_{em}$ = 757 nm), higher quantum yield, much larger two-photon absorption cross-section, and well-defined single-exponential kinetics. Mice were recovered from anesthesia and then head fixed under the microscope. The imaging session was started 30–60 min after injection, and lasted for up to 2 hr.

The acquisition protocol is illustrated in *Figure 1b*. We first recorded a CCD image of the mouse brain surface vasculature (*Figure 1d*) under green light illumination to guide the selection of a region of interest for the subsequent functional imaging. The two-photon microscopic measurements were collected at different X-Y planes, perpendicular to the optical axis (Z). The imaging planes were separated in depth by 50 µm. Starting from the cortical surface, up to 13 planes were interrogated per mouse, spanning the depth range of 600 µm. At each imaging depth, we first performed a raster scan of phosphorescence intensity, which revealed the locations of the micro-vessels within a 500 × 500 µm² field of view (FOV). Then, we manually selected the measurement locations inside all the vascular segments captured within the FOV. At each selected location, Oxyphor2P in the focal volume was excited with a 10-µs-long laser excitation at 950 nm gated by the EOM, followed by a 290-µs-long collection of the emitted phosphorescence. Typically, at each location, such 300-µs-long excitation/decay cycle was repeated 2000 times (0.6 s) to obtain an average phosphorescence decay with sufficient signal-to-noise ratio (SNR) for an accurate lifetime calculation. In some mice, longer acquisition (30,000 cycles; 9 s) was applied to enable measurement of EAT and temporal fluctuations of $PO_2$ and RBC flux (please see the following subsection for details).

## Measurement of RBC-$PO_2$, InterRBC-$PO_2$, EAT, and RBC flow in capillaries

In the capillaries in n = 7 mice, we repeated the 300-µs-long excitation/decay cycle 30,000 times, corresponding to a 9-s-long acquisition per capillary. In the capillaries in the other n = 8 mice, we

repeated the 300-μs-long excitation/decay cycle 2000 times, corresponding to a 0.6-s-long acquisition per capillary. At each imaging depth, all the capillaries that could be visually identified were selected for measurements. The 9-s-long acquisition allowed us to estimate RBC-PO$_2$, InterRBC-PO$_2$, EAT, and intracapillary PO$_2$ gradients, as well as the temporal fluctuations of RBC flux, speed and Mean-PO$_2$. Both 9-s-long and 0.6-s-long acquisitions were used to calculate Mean-PO$_2$, RBC flux, speed and line-density.

## Acquisition of microvascular angiograms

Microvascular angiograms were acquired by two-photon microscopic imaging of blood plasma labeled with dextran-conjugated Sulforhodamine-B (SRB) (0.15–0.2 ml at 5 % W/V in saline, R9379, Sigma Aldrich). The SRB solution was retro-orbitally injected into the blood stream under brief isoflurane anesthesia (1.5–2%, during ~2 min). The microvascular stacks were acquired within a 700 × 700 μm$^2$ FOV, centered over the same region of interest as for PO$_2$ imaging. In each mouse, the angiogram and intravascular PO$_2$ were acquired separately on different days in order not to stress the animals by the combined long imaging session.

## Calculation of PO$_2$

We rejected the initial 5-μs phosphorescence decay data after the 10-μs-long excitation gate, and used the remaining 285-μs decay to fit for the phosphorescence lifetime. The phosphorescence lifetime was calculated by fitting the average phosphorescence decay to a single-exponential decay function, using a standard non-linear least square minimization algorithm (*Finikova et al., 2008*; *Sakadžić et al., 2010*). The lifetime was converted to absolute PO$_2$ using a Stern-Volmer type calibration plot obtained in an independent oxygen titration experiment, conducted with the same 300-μs-long excitation/decay acquisition protocol as in our in vivo recordings.

## Calculation of capillary RBC flux, speed and line-density

The phosphorescence intensity was calculated by integrating the phosphorescence photon counts over each 300-μs-long excitation/decay cycle. Same as all the dendritic oxygen probes, Oxyphor2P is confined to blood plasma, but not permeates RBCs (*Lebedev et al., 2009*), the variations of the phosphorescence intensity recorded in capillaries encoded the passing of RBCs through the optical focus (*Lecoq et al., 2011*; *Parpaleix et al., 2013*) (*Figure 1b,c*). Following the previously described procedures (*Lecoq et al., 2011*; *Parpaleix et al., 2013*), the phosphorescence intensity time course was segmented using a standard thresholding method (*Otsu, 1979*). The segmentation of the phosphorescence intensity time courses was evaluated by the coefficient of determination (R$^2$) between the experimental and fitted time courses, and the data with R$^2$ <0.5 were rejected. A representative phosphorescence intensity time course is shown in *Figure 1c*, where a RBC and a blood-plasma-passage induced phosphorescence intensity transients are denoted by arrows. Subsequently, capillary RBC flux was calculated by counting the number of detected RBCs (i.e., valleys in the binary segmented curve) during the acquisition time. RBC line-density was estimated as the ratio of the combined time duration of all valleys to the total duration of the entire time course. Finally, RBC speed for each RBC-passage event was estimated as v = ø/Δt, where Δt is the time for the RBC to pass through the focal zone, and ø is RBC diameter, assumed to be 6 μm (*Unekawa et al., 2010*). For each capillary, the mean RBC speed was calculated by averaging over all RBC-passage events throughout the time course. Please note that the method for calculating RBC flow properties as described above is only valid for single-file-flow vessels, which are typically capillaries.

## Calculation of intracapillary PO$_2$ gradients, RBC-PO$_2$, InterRBC-PO$_2$ and EAT

To estimate the intracapillary PO$_2$ gradients in *Figure 6a*, the phosphorescence decays from the 9-s-long acquisition in each capillary were binned using 2-ms-wide bins, starting from the closest RBC center. The decays from the 9-s-long acquisition were subsequently averaged and PO$_2$ was calculated using the previously described procedures.

To estimate the intracapillary PO$_2$ gradients in *Figure 6—figure supplement 1*, the phosphorescence decays were grouped by their distance to the nearest RBC center, using 1-μm-wide bins. The distance to the nearest RBC center was estimated as v•Δt', where v was the RBC speed, and Δt' was

the time interval between the phosphorescence decay event and the center of the nearest RBC-passage (i.e., center of the valley). The decays from the 9-s-long acquisition were subsequently averaged and PO$_2$ was calculated as described previously.

With the same 9-s-long measurements, RBC-PO$_2$ was calculated with all the phosphorescence decays in the valleys (RBC-passages) in the segmented phosphorescence intensity time course (*Figure 1c*). InterRBC-PO$_2$ was calculated with the decays in the central 40% of the peaks (plasma). EAT was calculated as RBC-PO$_2$ - InterRBC-PO$_2$. Mean-PO$_2$ was calculated based on all the decays, regardless of their positions in the phosphorescence intensity time course. The calculation procedures were similar to what was described in *Lecoq et al. (2011)*; *Parpaleix et al. (2013)*; *Lyons et al. (2016)*.

## Calculation of SO$_2$ and depth-dependent OEF

The oxygen saturation of hemoglobin (SO$_2$) was computed based on PO$_2$ using the Hill equation with the parameters (h = 2.59, P$_{50}$ = 40.2 mmHg) specific for C57BL/6 mice (*Uchida et al., 1998*). Here, h is the Hill coefficient, and P$_{50}$ is the oxygen tension at which hemoglobin is 50% saturated. SO$_2$ in the penetrating arterioles and surfacing venules was calculated based on their Mean-PO$_2$. SO$_2$ in capillaries was calculated based on the RBC-PO$_2$ (*Lyons et al., 2016*; *Sakadžić et al., 2014*).

The depth-dependent OEF (DOEF), in a given cortical layer, was calculated as (SO$_{2,A}$–SO$_{2,V}$)/SO$_{2,A}$, where SO$_{2,A}$ and SO$_{2,V}$ represent the layer-specific SO$_2$ in the diving arterioles and surfacing venules, respectively. Therefore, DOEF in a certain layer measures the OEF accumulated downstream from that layer and, as a special case, DOEF in layer I represents the global OEF in the interrogated cortical tissue territory. Please note that DOEF in each layer depends on both blood flow and oxygen metabolism.

## Quantification of the temporal fluctuations of capillary Mean-PO$_2$, RBC flux, speed and line-density

The phosphorescence decays recorded during the 9-s-long acquisition (30,000 repetitions of the excitation/decay cycle) in each capillary were divided into 15 groups using 0.6 s bins (2000 repetitions for each bin). Here, averaging the phosphorescence decays with 0.6 s bins ensured that the blood flow and PO$_2$ fluctuations due to cardiac and respiratory cycles, as they occurred to awake mice, were averaged out. RBC flux, speed, line-density and Mean-PO$_2$ were calculated with the 2000 phosphorescence decays at each bin, yielding 15-point time courses of these four parameters for each assessed capillary. Subsequently, for each of the four parameters, we quantified the temporal fluctuation by computing the standard deviation (STD) and coefficient of variance (CV) from the 15-point data. Here, CV is defined as the ratio of STD to mean (*Golub and Pittman, 2005*).

## Identification of capillary branching order

Capillaries were typically identified starting one or two segments away from the diving arterioles and surfacing venules based on visually inspecting their morphology (e.g. smaller diameters and higher tortuosity) (*Sakadžić et al., 2014*). Their branching order indices were assigned by visually inspecting the microvascular angiograms. Starting immediately after the pre-capillary arteriole (PCA), capillary segments were counted in the direction of blood flow and indexed as Ai (i = 1, 2, 3, . . .). Analogously, starting immediately before the post-capillary venule (PCV), capillary segments were counted in the opposite direction of blood flow and indexed as Vi (i = 1, 2, 3, . . .). In the analysis, only the first three upstream capillary segments (A1-A3) and the last three downstream capillary segments (V1-V3) were considered, as visually inspecting and confirming higher branching orders of capillaries would be more challenging. In addition, most of the capillaries selected were within the central part of the FOV of the microvascular angiograms and in the cortical depth of ≤300 μm, which were due to the difficulty in tracking the capillaries close to the boundaries of the FOV and at greater depth. PCAs and PCVs were identified based on their PO$_2$ values and by visually inspecting their morphology, as well as by tracing the vessels up to the brain surface where we identified pial arterioles and venules based on their morphology and PO$_2$.

## Cortical layer-specific data analysis

The capillary RBC flow and $PO_2$ properties acquired in each animal were grouped into four groups based on the cortical depth: 0–100 µm, 100–320 µm, 320–450 µm, and 450–600 µm. These depth ranges approximately correspond to the cortical layers I, II/III, IV, and V, respectively, in the whisker barrel cortex in 3-month-old C57BL/6 mice (*Blinder et al., 2013*; *Lefort et al., 2009*). For each cortical layer in each mouse, the absolute values of capillary RBC flow and $PO_2$ properties were averaged, and the STD and CV computed. Subsequently, the measurements belonging to each cortical layer were averaged over mice.

## Rejection of motion artifacts

Data affected by mouse motion were rejected based on the signal generated by the accelerometer attached to the fabric underneath the mouse. We excluded from analysis the phosphorescence decays acquired within the time intervals determined by an empirically defined threshold of the accelerometer signal amplitude. In addition, visual inspection of the phosphorescence intensity traces acquired in the microvascular segments was also used to find and reject motion artifacts. This was achieved by looking for the sudden changes of phosphorescence intensity or loss of contrast between RBC and plasma passing through the focal volume. Finally, long episodes of motion were captured by the live-videos recorded by a CCD camera during acquisition. When motion occurred, the acquisition was manually stopped and the corresponding measurements were excluded from the analysis.

## Construction of the composite image

To construct composite images such as the one shown in *Figure 1g*, tubeness filtering (*Sato et al., 1998*) and intensity thresholding were applied to segment the two-photon angiograms into binary images. Subsequently, $PO_2$ measurements were spatially co-registered with the segmented angiogram. The three-dimensional composite image was created by color-coding the experimental $PO_2$ values in the corresponding vascular segments (shades of gray). The color-coding was performed by assigning the $PO_2$ (or Mean-$PO_2$ for capillary) value measured in the focal volume within a vascular segment to the whole segment. For some vascular segments without $PO_2$ measurements, such measurements were instead available for the segments joining them in each end. To such segments, we assigned the average $PO_2$ values of the connecting segments.

## Statistical analysis

Statistical comparisons were carried out using ANOVA or t-test (MATLAB, MathWorks Inc). p-Value less than 0.05 was considered statistically significant. Details about the statistical analysis and measurement information are provided in the figure legends and/or text, where relevant. Mean values and standard deviations of parameters needed to estimate the sample size were either assumed to be the same as those measured previously in anesthetized animals or assumed empirically. Since multiple parameters were measured in the same animals, sample size (i.e. n = 15 mice) was set based on anticipation that the most demanding one will be to detect 30% difference between the mean EAT values (coefficient of variance = 0.3, power = 0.8, $\alpha$ = 0.05).

## Data availability

All data generated or analyzed during this study are included in this paper and the supporting files.

## Acknowledgements

Support of the grants NS091230, MH111359, EB018464, NS092986, NS055104 and AA027097 from the National Institutes of Health, USA, is gratefully acknowledged.

## Additional information

### Funding

| Funder | Grant reference number | Author |
| --- | --- | --- |
| National Institutes of Health | NS091230 | Sava Sakadžić |
| National Institutes of Health | MH111359 | Anna Devor |
| National Institutes of Health | EB018464 | Sergei A Vinogradov |
| National Institutes of Health | NS092986 | Sergei A Vinogradov |
| National Institutes of Health | NS055104 | Sava Sakadžić |
| National Institutes of Health | AA027097 | Mohammad A Yaseen |

The funders had no role in study design, data collection and interpretation, or the decision to submit the work for publication.

### Author contributions

Baoqiang Li, Conceptualization, Data curation, Software, Formal analysis, Investigation, Visualization, Methodology, Writing—original draft, Writing—review and editing; Tatiana V Esipova, Ikbal Sencan, Buyin Fu, Investigation, Methodology; Kıvılcım Kılıç, Investigation, Methodology, Writing—original draft; Michele Desjardins, Frederic Lesage, Validation, Investigation, Methodology, Writing—review and editing; Mohammad Moeini, Software, Validation, Methodology, Writing—review and editing; Sreekanth Kura, Software, Visualization, Writing—original draft; Mohammad A Yaseen, Software, Validation, Methodology; Leif Østergaard, Validation, Writing—review and editing; Anna Devor, David A Boas, Sergei A Vinogradov, Resources, Funding acquisition, Validation, Investigation, Methodology, Writing—review and editing; Sava Sakadžić, Conceptualization, Resources, Software, Supervision, Funding acquisition, Validation, Investigation, Methodology, Project administration, Writing—review and editing

### Author ORCIDs

Baoqiang Li https://orcid.org/0000-0003-2992-3303
Mohammad A Yaseen https://orcid.org/0000-0002-4154-152X
Anna Devor http://orcid.org/0000-0002-5143-3960
Sergei A Vinogradov https://orcid.org/0000-0002-4649-5534
Sava Sakadžić https://orcid.org/0000-0001-6318-1193

### Ethics

Animal experimentation: All animal surgical and experimental procedures were conducted following the Guide for the Care and Use of Laboratory Animals and approved by the Massachusetts General Hospital Subcommittee on Research Animal Care (Protocol No.: 2007N000050).

### Decision letter and Author response

Decision letter https://doi.org/10.7554/eLife.42299.034
Author response https://doi.org/10.7554/eLife.42299.035

## Additional files

### Supplementary files

• Supplementary file 1. Measurement information for the main analysis in *Figures 2–7*.
DOI: https://doi.org/10.7554/eLife.42299.022

• Transparent reporting form
DOI: https://doi.org/10.7554/eLife.42299.023

## Data availability

All data generated or analyzed during this study are included in this paper and the supporting files.

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
