## [Decision Letter]

[Editors’ note: this article was originally rejected after discussions between the reviewers, but the authors were invited to resubmit after an appeal against the decision.]

Thank you for submitting your work entitled "Homogenization of capillary flow and oxygenation in deeper cortical layers correlates with increased oxygen extraction" for consideration by *eLife*. Your article has been reviewed by three peer reviewers, including Serge Charpak as the guest Reviewing Editor and Reviewer #1, and the evaluation has been overseen by a Senior Editor. The following individual involved in review of your submission has also agreed to reveal his identity: Andreas Linninger (Reviewer #2).

The reviewers have discussed the reviews with one another. All appreciated the quality of the data, however several analysis, hypothesis and statistical problems were raised, casting doubts on the article conclusion. As the amount of work required to improve the manuscript in a 2-month time period seems too important, the Reviewing and Senior Editors have taken the decision to reject the manuscript.

Please find below the detailed reviewer comments:

*Reviewer #1:*

Li's paper reports 2PLM measurements of PO2 in cortical vessels distributed from layer I-5, in awake mice. Using the new 2P phosphorescence probe PtTAPIP, synthesized by the group of S. Vinogradov (one of the co-authors) and which has an excellent 2PA cross section, the authors succeed in detecting all capillary parameters: mean-PO2, RBC flux and velocity and also erythrocyte-associated transients i.e. RBC PO2, Inter RBC-PO2. These measurements are reported for each layer of the cortex.

The work is interesting but lacks novelty and the analysis is not rigorously done. The Introduction requires a paragraph describing previous theoretical and experimental demonstrations of EATs and the scientific reasons for which the authors could not detected EATs previously. The community of 2PLM users is now expanding and it is important to mention all the flaws the initial labs working with this approach have been through. The introduction should properly describe previous 2PLM works reporting PO2 in mouse cortical layers. Surprisingly, the statistical tests are not adapted to the data preventing the interpretation of most comparisons. To conclude, the new dye is certainly a technical improvement, but the present manuscript requires major rewriting, analysis and does not reach the standards of *eLife*.

Technical comments:

Subsection “Calculation of capillary RBC flux, speed, and hematocrit”: The authors estimate RBC velocity as "v = ø/Δt, where Δt is the time for the RBC to pass through the focal zone, and ø is RBC diameter, assumed to be 6 μm (Unekawa et al., 2010). RBC diameter cannot assume to be 6 µm: in capillaries, RBC orientation and thus "RBC size" varies with capillary diameter, RBC density and velocity. As the shadow size in time, Δt, depends on the "RBC size", and thus velocity, it cannot be used to calculate velocity. All velocity measurements should be removed from the paper.

This also raises a problem in the way EATs are defined, as v.Δt is used to determine the distance to the nearest RBC center. I suggest that the authors reanalyze their data considering time and not distance to extract EATs.

Subsection “Calculation of capillary RBC flux, speed, and hematocrit”: The authors make the same mistake as Parpaleix et al., (2013) and Lyons et al., (2016) in the way they estimate hematocrit, which is normally a measure of blood volume percentage: "Hematocrit was estimated as the ratio of the combined duration of all valleys associated with the RBC passages to the duration of the entire time course." Because RBC elongation varies with velocity (see first comment), the hematocrit calculated in the paper will depend on velocity. The authors should name differently what is actually measured.

It is difficult to understand how RBC PO2 was determined. Could the authors elaborate on their approach: did they consider the first bin ("micron") or an average several bins to determine RBC PO2? Additionally, the use of the criterion of "the central 40% of the peaks in the binary segmented time course" to extract InterRBC-PO2 could be prone to error. Given that the InterRBC-PO2 is defined at the lowest value of plasma PO2 reached between the passage of RBCs, when the RBC flux is low, the use of the central 40% criterion could yield an accurate value of the this parameter, but is likely to become less accurate with increasing RBC flux, as the period when the plasma PO2 is at its minimum will be shorter, and the inflection will be sharper. The use of this criterion should be validated, comparing the InterRBC-PO2 value it provides with those values that are extracted from more restricted windows at greater distance from the RBCs.

All statistics are based on Student's t-test whereas in almost all comparisons, ANOVA test with multiple comparison analysis should be used.

Subsection “Calculation of capillary PO2 gradients and EATs” and subsection “Accumulated oxygen extraction fraction increases in deeper cortical layers” The authors estimate layer-specific oxygen extraction fraction by comparing the PO2 in arterioles and venules at a given depth. This approach tacitly assumes that a layer-specific network of microvessels connects arterioles and venules in this layer, and that all RBCs which leave arterioles in the given FOV and layer flow into venules in that same field of view and layer. The authors should cite which theoretical and experimental papers support the hypothesis. In addition, as they have the tools to reconstruct easily the vascular angiograms, they should trace a series of pathways from the pre-capillary arteriole to the post-capillary venule in order to verify the hypothesis, which is key for the paper. They could potentially limit their investigation to layer I and IV pathways, which are likely to differ.

Is the mouse's head rotated around the rostro-caudal axis during the 2PLM sessions? If not (and unless the objective lens can be inclined, thought this doesn't appear to be the case in Supplementary Figure 1(b)), due to the inclination of the cortical surface at the coordinates of the barrel field it is probable that there is a discrepancy between the reported measurement depths and the true depth of measurement in the cortex. As illustration, the approximately 40-45 degree inclination of the cortex relative to the horizontal at the level of the S1 barrel field (as per Paxinos and Watson), would mean that the maximum reported imaging depth of 600 micros would infact correspond to a true cortical depth of approximately 460 microns. Thus, the assignation of the measurements to different layers will be compromised. Conversely, if the mouse head is rotated, could the authors note this and comment on the likely effect on the comfort and behavioural state of the mouse during the imaging sessions.

Biological:

3D projections of PO2 maps (Figure 1G) are made with acquisitions lasting only 0.6 seconds (2000 decays). This is really short as PO2 varies with time. It is clearly shown in Figure 5. PO2 may change by more than a factor 2 within 9 s. Therefore, PO2 maps using such brief acquisitions will change dramatically from one acquisition to the other. What is the scientific value of such 3D map? In addition, it artificially increases the number of vessels imaged.

If this approach was also used to build Figure 2, it strongly decreases its significance.

In the absence of proper statistics, Figure 3, Figure 4, Figure 5, Figure 6 and Figure 7 cannot be interpreted.

In their 2014 paper, the authors reported that PO2 decreases with the capillary order. Could the author verify if their findings hold true in the awake mouse? This would require a simple analysis regrouping A1, A2 and A3 capillaries.

Given the concerns raised above about the definition of RBC-PO2 in capillaries and the method for estimation of OEF by comparing arterioles and venules (and indeed A1-A3 and V1-V3 capillaires) in the same layer, it follows that the estimation of the relative contribution of different vascular compartments and the related conclusions (Subsection “Low oxygen extraction along the superficial capillary paths contributes to an increase in the mean venular SO2 towards cortical surface”) are questionable.

*Reviewer #2:*

The article presents novel data on hemodynamic states in the different layers of the whisker barrel cortex of awake mice. in vivo data are acquired using a novel oxygen probing technique with two-photon phosphorescence lifetime microscopy. Specifically, steady averaged oxygen pressures and red blood cell (RBC) flux counts were acquired. In addition, temporal variations within a 9s observation window were collected. A detailed statistical analysis of the layer dependence of oxygenation and RBC fluxes is presented. The main conclusions include the experimental observation of depth dependence of oxygen extraction, and a reduction in the RBC flux variability with increasing cortical depth. This study presents an invaluable experimental body of work that will help to elucidate oxygen extraction mechanisms in the mouse cortex and create in vivo data for mechanistic models to explain the physiochemical principles that drive and control cerebral blood flow and metabolism. This is a significant piece of work which I highly recommend for publication after modifications listed below:

Technical comments:

Introduction. The EAT is used to characterize the heterogeneity of the oxygen extraction within capillaries. What is the physical rationale for correlating oxygen point measurements to intracapillary resistance to oxygen delivery? Are these measurements for erythrocyte bound versus unbound oxygen in plasma? How is this EAT related to oxygen saturation?

It appears that measurements of RBC fluxes (RBC counts) assume that signals are generated from individual RBCs that are sharply separated. Figure 1C shows signals with different durations (and intensity). Is it possible that two aligned RBCs (two or more RBCs in file) may cause a longer signal that is indistinguishable from a single slow RBC? How would this affect the analysis of RBC fluxes?

In the Introduction, the authors cite the next generation of biophysical models, which with the exception of Gagnon's 2016 paper do not correspond to the anatomical detail presented in this study. Layer specific oxygen consumption has been predicted by recent biophysical models that match the detailed anatomical scope and three-dimensional resolution of the proposed experimental study. For example, three dimensional predictions of depth dependent oxygen gradients in mouse are given in Gould et al., 2017. Biophysical models to predict depth dependent oxygen gradients in humans are presented in Linninger et al., 2013 and Gould and Linninger, 2015. These studies have been successful in solving oxygen delivery to brain tissue coupled with biphasic blood flow in a realistic cortical microanatomy and are therefore perfectly aligned with the scope of the current study. These advancements should therefore be incorporated and discussed.

Subsection “Accumulated oxygen extraction fraction increases in deeper cortical layers”. The data in Figure 2 seem to refer to steady magnitudes or levels, the use of the word "amplitudes" seems ambiguous.

Subsection “Capillary RBC flux and oxygenation homogenize in deeper cortical layers”. The prediction that homogenization of RBC flow velocity enhances the oxygen extraction is made based on a single segment analysis in the cited biophysical model. Here, experiments are presented for a three-dimensional vascular network. I am uncertain whether the two scopes are really "in line" with each other as suggested in the current text.

Figure 5. The trend line in Figure 5 does not intersect at zero. What does the non-zero intercept of roughly 16-18mmHg pO2 mean? Should the trend have a zero intercept?

Subsection “Capillary RBC flux and oxygenation homogenize in deeper cortical layers”. The number of segments in layer IV and V have only half the segments than layer I-III. Could this affect the statistics? If not, this observation should be listed in the limitations.

The assignment of upstream capillaries A1-A3 and downstream capillaries V1-V3 was done manually. Is it hard to imagine that roughly hundred segments (=according to subsection “Low oxygen extraction along the superficial capillary paths contributes to an increase in the mean venular SO2 towards cortical surface”, the analysis included 97 segments for all mice specimen) were identified by hand without the aid of image filters (e.g. Blinder et al., 2013; Hsu et al., 2017) in combination with automatic segmentation.

In subsection “Low oxygen extraction along the superficial capillary paths contributes to an increase in the mean venular SO2 towards cortical surface” it is argued that the manual method provides statistical power. It is not clear what statistical power is referred to? I suspect that the results could come out quite differently, if labels were assigned differently. Is there any data on operator dependence of the labeling method? I recommend to consider writing the discussion section more cautiously to reflect that results are based on an operator dependent method with high uncertainty in segmentation, and that major results would not be affected by uncertainty associated with operator dependence in upstream and downstream labeling. This point is optional to the discretion of the authors.

Subsection “Low oxygen extraction along the superficial capillary paths contributes to an increase in the mean venular SO2 towards cortical surface”. The hematocrit changes were reported between upstream (A1-A3) and downstream capillaries (V1-V3). Why did the study not explore layer dependence or branch hierarchy dependence (Strahler order analysis) of hematocrit?

Discussion section. The RBC flux is equal to the product of bulk flow rate and hematocrit (volume fraction). How should we understand the variance reduction in RBC flux without any relation to variations in hematocrit, since the quantities are directly related?

Discussion section. The wide variability of hemodynamic states (high heterogeneity of capillary flow and oxygenation) was predicted previously in a biophysical model by Gould et al.,.2017. It would be helpful if the experimental results could be aligned with already completed theoretical work that aims at addressing the same points as those that are here so elegantly presented experimentally.

In the same vein, it is worth mentioning that a main finding of the reduced variance of RBC fluxes was just recently predicted covering the whisker barrel mouse cortex with extension to the entire MCA territory (Hartung et al., 2018). The experiments confirm several findings so that the predictive model results are highly relevant for this study and should be discussed.

*Reviewer #3:*

The work addresses highly relevant questions (depth-dependent difference, homogeneity/heterogeneity of microvascular flow and oxygenation). The amount of measurements performed is significant and the capabilities of the new oxygen probe PtTAPIP to measure a variety of blood flow characteristics in large cortical depths is nicely demonstrated. However, we have major concerns regarding some of the analysis performed and some of the conclusions drawn (listed below). Furthermore, there is a significant shortcoming in referencing earlier work. It remains unclear, where the current work goes beyond what was published before.

Essential revisions:

1) Calculation of capillary RBC flux, speed and hematocrit (subsection “Calculation of capillary RBC flux, speed, and hematocrit”):

As the authors are well aware, the methods presented to compute RBC flux, RBC speed and hematocrit are only valid in vessels where single file flow persists. The authors write that those measurements were performed in "capillaries". However, it remains unclear how capillaries were identified as capillaries. Were diameter measurements performed? As the RBC diameter of the mouse is 6 µm, single file flow can only be expected for vessel diameters < 6 µm. This aspect is key for the validity of the results. We propose that the authors come up with a table describing the measured vessels (type, depth, number) for clarity.

1a) Additional Concern regarding velocity calculations (subsection “Calculation of capillary RBC flux, speed, and hematocrit”):

Characteristics Mouse RBC: Diameter 6 µm, Volume: 45.5 µm3 (Windberger et al., 2013), Thickness: 1.6 µm.

Assuming that each RBC is 6 µm is not correct. Especially, in larger vessels the RBC orientation makes a significant difference in the proposed velocity calculation (by a factor of ~4).

For vessel diameters <5.5 µm the applied velocity calculation is more suitable, because here RBCs need to squeeze into the vessel. However, also here we need to account for changes in RBC length depending on the vessel diameter.

In summary, the presented approach gives at best a rough estimate of the RBC velocity and it remains unclear, why the authors have not chosen line scans that would be so much more appropriate.

1b) Additional Concern regarding hematocrit calculations (subsection “Calculation of capillary RBC flux, speed, and hematocrit”):

The hematocrit computation is based on the assumption that plasma and RBCs have the same velocity. However, RBCs travel on average faster than plasma (Fahraeus effect). This velocity difference can cause an underestimation of hematocrit. Even so this approach is used frequently this assumption should be described and the term "line-density" should be used instead of "hematocrit".

1c) Additional Concern regarding flux and velocity calculation (subsection “Calculation of capillary RBC flux, speed, and hematocrit”):

For the flux and velocity computation it is crucial that individual RBCs can be distinguished from another. Can this be guaranteed for all hematocrit levels? Some discussion/comments should be added.

1d) Selection of vessels for analysis (Subsection “Capillary RBC flux and oxygenation homogenize in deeper cortical layers”, subsection “Imaging of EATs and capillary RBC flow”):

The selection criterion for the vessels that have been chosen to analyse flow properties are not described. Moreover, it remains unclear why the number of vessels investigated per layer differs so significantly (400, 356, 118, 104).

2) Quantification of the temporal fluctuations of capillary Mean-PO2, RBC flux, speed and hematocrit (subsection “Quantification of the temporal fluctuations of capillary Mean-PO2, RBC flux, speed, and hematocrit”, Discussion section):

On the one hand, the chosen measurement time is rather short for a good averaging of the readouts. On the other hand, an averaging interval of 0.6 s is too large to resolve fluctuations in the capillary bed. An RBC with an average velocity of 1mm/s would travel 600 µm during that time. So, for the overall statistics, longer periods should be measured, but for describing the fluctuations over time, those segments should be split up into smaller than 0.6 s intervals, so that the fluctuations are dampened too much due to the averaging.

The arguments given in the discussion for the chosen interval seem to be motivated by dynamics on a larger scale but not by microvascular fluctuations. Why is the rate of oxygen consumption relevant for the capturing microvascular flow dynamics?

Alternatively, it has to be described more clearly that only fluctuations on the time scale of seconds are analysed and that faster fluctuations persist but are not of interest in the current study.

3) Analysis of EATs:

Intracapillary resistance of oxygen transport to tissue decreases in deeper cortical layers (subsection “Intracapillary resistance of oxygen transport to tissue decreases in deeper cortical layers”) and Low oxygen extraction along the superficial capillary paths contributes to an increase in the mean venular SO2 towards cortical surface (subsection “Low oxygen extraction along the superficial capillary paths contributes to an increase in the mean venular SO2 towards cortical surface”):

In order to understand the observed EAT trends, it is crucial to also analyse RBC-PO2 and interRBC-PO2. This aspect has been neglected for most EAT results. As such, possible reasons for the observed EAT drop in Figure 6C and the EAT increase in V1-V3 segments have not been analysed sufficiently (Based on Figure 3—figure supplement 1ARBC-PO2 does not drop? Thus interRBC-PO2 should rise for deeper layers to explain the EAT drop? How can a rise in RBC-PO2 and interRBC-PO2 for deeper cortical layers be explained? A higher interRBC-PO2 would however suggest a "higher intercapillary resistance to oxygen transport" instead of a lower one.)

Moreover, the EAT strongly depends on the distance of the capillary to the arteriole/venule (see Figure 7C). This impact should be discussed, e.g. how do you make sure that your depth-dependent EAT average is not affected by a larger number capillaries close to arterioles?

In an earlier work by the same authors, they did not observe EATs. The authors should discuss in detail where this discrepancy originates from.

4) Depth-dependent oxygen extraction (subsection “Accumulated oxygen extraction fraction increases in deeper cortical layers”, Discussion section):

We generally agree with the idea to use the difference in oxygen saturation as an indicator of oxygen extraction. However, as also shown in this work here, many factors have an impact on the oxygen saturation. Thus, saturation difference is not always equal to oxygen extraction. For example, higher blood flow in layer I leads to a higher oxygen availability in layer I and thus to higher SO2 values in the venules, even if the oxygen extraction is constant over depth.

"Benefitting from the improved sensitivity of the new oxygen probe, we were able to measure intracapillary longitudinal PO2 gradients in a larger number of capillaries…" why does the application of the improved probe help to increase the number of vessels measured?

[Editors’ note: what now follows is the decision letter after the authors submitted for further consideration.]

Thank you for submitting your article "More homogeneous capillary flow and oxygenation in deeper cortical layers correlate with increased oxygen extraction" for consideration by *eLife*. Your article has been reviewed by three peer reviewers, including Serge Charpak as the guest Reviewing Editor and Reviewer #1, and the evaluation has been overseen by Timothy Behrens as the Senior Editor. The following individual involved in review of your submission has also agreed to reveal his identity: Andreas Linninger (Reviewer #2).

The reviewers have discussed the reviews with one another and the Reviewing editor has drafted this decision to help you prepare a revised submission. All reviewers believe that the work is interesting. They however raised a number of questions that I ask you to address and which would clarify the manuscript. No new data is required. I have summarized the key questions:

1) The demonstration that RBC size can be estimated from the 2PLM measurements is still not convincing. The way you analyzed your new data is not informative: Comparing the mean RBC values from 58 capillaries measured with Line-Scan and Point-Scan (Author response image 2) does not solve the problem, in particular as the variability is large. Please show the plot in Author response image 2, with lines joining each values, capillary per capillary. It is important as Author response image 2 already seem to demonstrate that RBC longitudinal size varies a lot. The 3 reviewers believe that if your new analysis reveals that RBC size cannot be accurately estimated, all RBC velocity data should be removed from the paper and the resulting uncertainty of RBC PO2 measurements be carefully addressed in the discussion.

2) Please address in the discussion the issue that a saturation difference may depend on several factors.

3) Add the plot for interRBC-PO2 over depth to the supplementary figures.

4) Add in the Impact Statement that the measurements are done at the steady state.

The reviewers have added some comments (see the full reviews attached below) to which you could briefly answer.

*Reviewer #1:*

The authors have made new experiments, new analysis and the text has improved. Previous works are now better described and discussed. The statistics are more appropriate.

The new experiments and analysis done in response to comments #2 and #3 are interesting but they do not fully address the issues raised. This concerns primarily RBC velocity measurements. The good news is that it could be easily done with simple analysis without any further experiment.

The authors initially used the assumption of an identical average size of RBCs to base other calculations, in particular RBC speed. They now show new experiments which results are summarized in several "R1" plots. The plots show that RBC longitudinal size is very variable, increasing very significantly with RBC speed and spreading from 3 to 14 µm (R1C1261 c plot (<30% RBC line-density)). This clearly indicates that RBC shadows measured during PO2 acquisitions cannot be used to estimate RBC speed. Note that the authors should give the "n" for each of the groups of capillaries in all the new "R1" plots.

This variability of RBC size is unfortunately masked in the R1C2-2 plot. This plot needs to show paired measurements in specific capillaries, the question being whether or not similar values of RBC velocity are recorded by the two methods when (presumably) similar real RBC velocities are measured. Showing population averages for the 58 capillaries assessed occludes the similarity/difference on a capillary by capillary basis. The important point here is that all the data are already acquired, for capillaries of known diameters. So the authors could now easily compare RBC values for paired measurements (with the 2 methods) in capillaries of different diameters. Please indicate the "n" for each of the groups of capillaries and plot, for each group, all RBC speed values measured with the 2 methods. I suspect that the authors will find differences with the two techniques. This would invalidate all RBC measurements obtained with point scan PO2 acquisitions.

It is important that the authors clarify the point as fluctuations of RBC shadows, whether measured in distance or time from the center of the shadow, will modify the RBC PO2 value. In fact, it will decrease RBC PO2 and thus the EAT amplitude. Could the authors measure the decays as a function of time, but after an alignment at the RBC border? This would solve the problem.

*Reviewer #2:*

The authors have addressed all previous concerns and submitted a more concise revised manuscript. Well done.

*Reviewer #4:*

To the editors:

Overall the authors improved their manuscript by clarifying various methodological issues and by providing a more detailed introduction and discussion. Nonetheless, some major issues remain. In my opinion the evidence for some of the major results is not strong enough or to put it differently the claims are currently too strong for the presented results. To be more precise: (1) I am still not convinced by the RBC speed calculation (comment #1a). (2) Oxygen extraction and saturation difference are not necessarily equivalent (comment #4). (3) The analysis of the depth dependent EATs should be more rigorous (comment #3). Nonetheless, I believe that this work is relevant and builds onto a large body of experimental work. It can be further improved by a more rigorous analysis of the available data and a more concise discussion of the uncertainties in the presented results. However, this may require that some of the conclusions of the manuscript are slightly weakened/adapted. More details on the major issues are provided in the detailed reply, which follows below.

I thank the authors for the additional explanations and the adjustments made. Below I list the comments where additional clarification is necessary in order to answer the initial question. Only the comments where further adjustments are necessary are listed. The ones where major issues remain are highlighted and positioned at the beginning.

I also read the comments of reviewers 1 and 2 as well as the subsequent changes to the manuscript. As some of the points raised by the other reviewers are very relevant, I added an additional comment for some of them.

The additional studies are an important step to judge the accuracy of the RBC velocity measurements.

However, the following major concerns remain:

- Author response image 2 scatter plot that directly compares the RBC velocity from the line-scan and the point measurement would be more appropriate. Additionally, the average relative difference between the two measurements should be provided.

- Author response image 7: The results clearly show that there is significant variability in the longitudinal RBC size (CV = 0.4, longitudinal sizes ranging from 4-14 µm). Moreover, the RBC longitudinal RBC size is correlated with the RBC speed and the RBC Line-Density. It is impossible to estimate the impact of these dependencies on the presented RBC speed results.

I thank the authors for the additional explanations regarding IVR and I believe that the changes in the discussion are very valuable. However, some aspects of my initial comment have not been addressed.

The EAT is computed from the RBC-PO2 and the interRBC-PO2. As such I believe that EAT, RBC-PO2 and interRBC-PO2 should always be analysed and discussed hand in hand. Thus, it would be valuable for the manuscript to add the plot for the interRBC-PO2 over depth to the supplementary figures and to discuss RBC-PO2 and interRBC-PO2. Moreover, I suggest to add and discuss the EAT STD & CV plots over depth as it is done for all other quantities.

As stated in my original comment it is a surprising results that RBC-PO2 increases over depth and I don't know which mechanism could explain this increases (Figure 3—figure supplement 1). The same holds for interRBC-PO2 (which has to increase more than RBC PO2 in order to explain the EAT drop over depth). I do not ask for additional experiments, but I believe that is important to discuss these trends.

In my initial question I asked how the author's ensured that the depth-dependent differences are not affect by the position of the chosen capillaries along the capillary pathway or to put it differently how the capillaries were chosen over depth to guarantee an equal distribution of "upstream" and "downstream" capillaries. I kindly ask the author to describe if this has been considered in some way? If it has not been considered the possible impact on the depth dependent results should be discussed.

The EAT drop over depth is one of the major results of this manuscript and as the authors state in the discussion "EAT measurements are typically much noisier" (Discussion section). Consequently, I believe that the available data should be analysed as rigorously as possible.

I disagree with the given explanations why RBC flux in layer IV is supposed to be higher than in layer I. What matters here is not the average RBC flux and the higher capillary density per layer but the flux into the capillary bed per layer or to put it differently the flux out of the diving arterioles per layer.

The given arguments connecting average RBC flux and capillary density are not plausible. I try to explain this with a simplified example. Imaging two similar tissue volumes: One with a single vessel and flow rate q1 through that vessel. The second one also has an inflow rate of q1 but the vessel splits in two vessels. In both cases the inflow (and thus the oxygen availability) per tissue volume is the same. The vessel density is however higher in the second one. The authors now argue that the macroscopic flow rate would be larger in the second example, which is not true.

Of course, higher vessel density, i.e. more flow pathways, might have an effect on the overall flow rate. However, many open questions remain regarding these issues and simply relating RBC flux and capillary density to estimate the overall flow is not correct.

The referenced figures (Suppl. Figure 7b in Sakadazic et al., 2014 and Figure 2c in Gould et al., 2017) show the number of capillary segments, which is not the same as vessel density.

In the original work from Blinder et al., 2013 Figure 2c the capillary density increased from ~4% in layer I to ~5% in layer IV, which is an increase by ~20% but not by 50%.

Taken together, my initial question remains, i.e. the depth-dependent blood flow/oxygen availability has a strong impact on the actual oxygen extraction per layer. This should be discussed properly. Maybe it would good to change the variable name to depth-dependent saturation difference or comparable, because the term oxygen extraction seems to be misleading.

I believe the impact statement should be improved. "Homogenization", "Mechanism" and "adapts" suggest that the presented study looks at active mechanisms or dynamic changes. However, the work is a detailed description of the steady state flow and oxygen distribution.

[Editors’ note: further revisions were suggested before acceptance, as shown below.]

Thank you for resubmitting your work entitled "More homogeneous capillary flow and oxygenation in deeper cortical layers correlate with increased oxygen extraction" for further consideration at *eLife*. Your revised version has been discussed by the peer reviewers that raised some issues about your previous version and overseen by Serge Charpak as the guest Reviewing Editor and Timothy Behrens as the Senior Editor.

All acknowledged your efforts in responding to their comments. Most of your responses are satisfactory but some reviewers raised the concerns that your manuscript does not reflect at all the intense and fruitful discussion that occurred during the reviewing process. As the Reviewing Editor, I am pleased to inform you that your work is suitable for publication in *eLife*, providing that you include in the manuscript your responses to some of the questions/responses raised during the process of reviewing (see below). Note that most controversial points have been discarded. This will not take you more than a couple of hours and I will be pleased to address your revised version to the production department.

Please add in the manuscript:

Measurements of RBC velocity:

Several reviewers are still not fully convinced but accepted that the data are included in supplementary figures, providing that you add your work done to estimate the RBC size. The sole sentence line 145 (RBC speed calculation was model based …) and the discussion on RBC size are not fully satisfactory.

To end the controversy, I propose the following:

1) Subsection “Oxygen extraction fraction increases in the deeper cortical layers”: substitute the sentence by something like: "Note that as the instantaneous RBC shadow varies with both RBC speed, position and vessel size (see Figure 1—figure supplement 1), RBC speed calculation was model-based by assuming a constant RBC size (6µ)(Unekawaet al., 2010).

2) In Figure 1—figure supplement 1 (it will replace the current supplementary figure 1 which is not informative) please add the following plots which are interesting, justify the model based-choice and explain the problem of point measurements to estimate the RBC speed:

Add the plot from your summary comment (Title comparison between the RBC speed measurements by the line-scan and point-scan method.)

Add the new Author response image 7 plot (title Correlation between the line-scan and point-scan RBC-speed values in the capillaries having the diameter of 2-3 μm (left panel) and 3-5 μm (right panel)).

Add the plot Author response image 1 from your former response (Title a-c. RBC longitudinal size vs. capillary diameter, RBC speed, and line-density, respectively.)

3) Add the comments on these findings in the Discussion section.

Smaller EATs in layer IV:

Include in the discussion your detailed responses to the following points:

The increase in interRBC-PO2 and RBC-PO2 over depth, which is a very surprising result, as generally the most saturated RBCs enter the vasculature at the surface.

The impact of the sampling of "upstream" and "downstream" vessels and their distribution over depth.

Saturation difference/oxygen extraction:

As oxygen extraction, saturation difference and total blood flow are related quantities, it is important to add this information at two locations:

- where the depth-dependent oxygen extraction fraction is introduced (subsection “Oxygen extraction fraction increases in the deeper cortical layers”) and

- where the calculation of the depth-dependent OEF is described (subsection “Calculation of SO2 and depth-dependent OEF.”).

---

## [Author Response]

[Editors’ note: the author responses to the first round of peer review follow.]

The reviewers have discussed the reviews with one another. All appreciated the quality of the data, however several analysis, hypothesis and statistical problems were raised, casting doubts on the article conclusion. As the amount of work required to improve the manuscript in a 2-month time period seems too important, the Reviewing and Senior Editors have taken the decision to reject the manuscript.

Thank you for giving our manuscript an opportunity to be reviewed by three experts in the field. They raised several important questions and provided valuable suggestions how to improve the manuscript. While we understand the reasons behind this decision, we would like to reassure you that we can fully address the comments within the 2-month time period.

Please find below the detailed reviewer comments:

Reviewer #1:

Li's paper reports 2PLM measurements of PO2 in cortical vessels distributed from layer I-5, in awake mice. Using the new 2P phosphorescence probe PtTAPIP, synthesized by the group of S. Vinogradov (one of the co-authors) and which has an excellent 2PA cross section, the authors succeed in detecting all capillary parameters: mean-PO2, RBC flux and velocity and also erythrocyte-associated transients i.e. RBC PO2, Inter RBC-PO2. These measurements are reported for each layer of the cortex.The work is interesting but lacks novelty and the analysis is not rigorously done. The Introduction requires a paragraph describing previous theoretical and experimental demonstrations of EATs and the scientific reasons for which the authors could not detected EATs previously. The community of 2PLM users is now expanding and it is important to mention all the flaws the initial labs working with this approach have been through. The introduction should properly describe previous 2PLM works reporting PO2 in mouse cortical layers. Surprisingly, the statistical tests are not adapted to the data preventing the interpretation of most comparisons. To conclude, the new dye is certainly a technical improvement, but the present manuscript requires major rewriting, analysis and does not reach the standards of eLife.

We added the following text to the Introduction to include more details about the previous EATs modeling and measuring:

“The development of 2PLM of oxygen also enabled measurements of erythrocyte-associated transients (EATs) in cortical capillaries (Lecoq et al., 2011). EATs were first theoretically predicted by Hellums (Hellums, 1977) and extensively investigated over the last four decades using analytical and numerical approaches (Federspiel and Popel, 1986; Hellums, 1977; Lücker et al., 2015, 2017; Popel, 1989). Originally, they were experimentally observed in peripheral capillaries (Barker et al., 2007; Golub and Pittman, 2005), but the full confirmation within the more challenging three-dimensional cortical capillary network was made possible only recently with advent of 2PLM (Lecoq et al., 2011; Lyons et al., 2016; Parpaleix et al., 2013). Since EATs are tightly related to the intravascular resistance to oxygen transport to tissue, their direct measurements are critical for better understanding of the oxygen delivery through the capillary network. However, the dependence of EATs on cortical layer has not been fully explored.”

We respectfully disagree with the comment about the novelty. Our measurements not only probed deeper cortical layers than previously reported, but, importantly, we have found evidence that several parameters relevant for O_2_ transport to tissue are more homogeneous in deeper, presumably more energy demanding cortical layers, at rest. This has been proposed as a mechanism that facilitates O_2_ transport to tissue by several recent theoretical works. In addition, several new types of measurements in awake mice are presented in our manuscript, including O_2_ extraction in different cortical layers, arteriolar contribution to O_2_ delivery, and correlations between the fluctuations of capillary blood flow parameters. Having said that, we do agree with the reviewer that Introduction and Discussion section should be updated to better clarify these points as well as to provide better referencing and comparison with the previous findings. For the related changes in the manuscript text, please see our response to comment #7 by reviewer 3, who had similar requests for updating the text.

In the revised manuscript we applied ANOVA test to all the data whenever appropriate. The new statistical analysis did not change most important conclusions from the original submission. We revised the text in the Method section, subsection Statistical analysis, and it now reads:

“Statistical comparisons were carried out using ANOVA or t-test (MATLAB, MathWorks Inc). P value less than 0.05 was considered statistically significant. Details about the statistical analysis and measurement information are provided in the figure legends and/or text, where relevant.”

We agree that it is helpful to provide more details about the differences between previous EATs measurements by using the same imaging technology. The following sentences were added to Discussion section:

“We would also like to mention that some discrepancy exists between the magnitudes of EATs measured using 2PLM by different groups. For example, rather large EATs were reported in (Lecoq et al., 2011; Lyons et al., 2016; Parpaleix et al., 2013) with the PtP-C343 probe. Much smaller EATs (only a few mmHg) were reported in (Sakadžić et al., 2014) using the same probe, and moderate EATs are reported in the present work based on the measurements using the new probe – Oxyphor2P. It should be mentioned that the data underlying the EATs measurements are typically much noisier than those used to derive mean intravascular and tissue PO2's, since to quantify EATs the signals have to be split into multiple bins at different spatial or temporal distances from RBCs. In addition, the previously used probe PtP-C343 has intrinsically non-single-exponential phosphorescence decay and much lower emission quantum yield. These limitations, in combination with potentially slightly different implementations of acquisition protocols and algorithms for fitting the phosphorescence decay data, are likely to be the dominant factor contributing to the differences in the reported EATs magnitudes. By using Oxyphor2P, which has much stronger signal and much better defined single-exponential decay, making data analysis much more robust, here we greatly reduced uncertainty in the measurements of EATs, which was inherent to the previous probe PtP-C343. Nevertheless, we still would like to emphasize the importance of aligning the acquisition and data analysis protocols across the labs as well as using the same data acquisition protocols during measurement as used for probe calibration.”

Technical comments:Subsection “Calculation of capillary RBC flux, speed, and hematocrit”: The authors estimate RBC velocity as "v = ø/Δt, where Δt is the time for the RBC to pass through the focal zone, and ø is RBC diameter, assumed to be 6 μm (Unekawa et al., 2010). RBC diameter cannot assume to be 6 µm: in capillaries, RBC orientation and thus "RBC size" varies with capillary diameter, RBC density and velocity. As the shadow size in time, Δt, depends on the "RBC size", and thus velocity, it cannot be used to calculate velocity. All velocity measurements should be removed from the paper.This also raises a problem in the way EATs are defined, as v.Δt is used to determine the distance to the nearest RBC center. I suggest that the authors reanalyze their data considering time and not distance to extract EATs.In addition, in response to the Appeal Letter, the editors provided suggestions closely related to the above technical comments.“The Reviewing Editor and the Senior Editor have carefully read your appeal letter and agreed that you could send a revised version of your manuscript provided that you succeed in responding to all the points raised by the reviewers. Still, the reviewing editor stresses that in view of the strong concerns raised about your estimation of RBC velocity, you should perform a series of experiments that could be easily done and would test the validity of your hypothesis: you should select several groups of capillaries, with large (5-6 m) and small (2-3 m) diameters, with low (0.2-0.5 mm/s) and high (1-1.5 mm/s) velocities (for both small and large capillaries), and measure the instantaneous longitudinal size of RBC shadows using the line scan approach. This would allow you to verify the extent to which the RBC shadow size varies with time, as well as with the capillary type and velocity (and density, if possible) (See the study in rats by Chaigneau et al., 2003). Depending of your findings, i.e. the level of RBC "size" stability, you should be able to conclude whether your estimation of RBC velocity and its use to detect EATs will remain valid. You could also perform paired point-scan based measurements and line scan measurements in capillaries with stable flow and directly compare whether a similar value of mean RBC velocity is obtained. Note that this latter approach would not tell whether fast (instantaneous) changes of RBC shadow are too frequent to allow the use of RBC velocity to detect EATs.”

We agree with reviewer that the point-scan based RBC speed measurement (also referred to as the *RBC-passage* method) has limitation imposed by assuming the constant RBC diameter and thank the reviewer for suggesting the appropriate experiments to better clarify this issue. Below, we followed the reviewer’s suggestion to validate the point-scan-based RBC speed estimation with additional measurements.

We performed line-scan measurements in two awake C57BL/6 mice (3-5 months old, female, 20-25 g, Charles River Laboratories). The cranial window was prepared following the same protocol as described in the manuscript. We injected dextran-conjugated Sulforhodamine-B (0.1 ml at 5% W/V in saline, Σ R9379) to label the blood plasma. The RBC speed was measured in 58 capillaries by the line-scan technique (2-s-long acquisition in each capillary with the line-scan frequency 2000 Hz). In addition, we extracted the fluorescence intensity time courses from the same parallel line-scan images, and then the RBC flux and speed were calculated with the procedures described in the manuscript. For each capillary, we also estimated the capillary diameter by fitting the transversal intensity profile to a Gaussian. The diameters were calculated as the full width at half maximum of the Gaussian profiles. Data with R^2^<0.5 were not considered for analysis. The RBC line density and RBC longitudinal size were calculated by following the procedures described in (Chaigneau et al., 2003).

By averaging over all RBCs in each capillary and then across all 58 capillaries, we obtained the mean RBC longitudinal size (6.9 ± 3.0 µm; Mean ± STD). We also quantified the variation of the longitudinal sizes of RBCs passing through the capillary during 2 s. First, we computed for each capillary the standard deviation (STD) and coefficient of variance (CV) of the longitudinal sizes of all the RBCs measured during the 2-s-long acquisition, where CV was calculated as the ratio of STD to mean RBC longitudinal size within the capillary. Then, we averaged obtained STD and CV values across the 58 capillaries (mean STD = 2.3 ± 0.6 µm; mean CV = 0.4 ± 0.1).

**Author response image 1. respfig1:** RBC longitudinal size vs. capillary diameter, RBC speed, and line-density, respectively. The measurements were performed in 58 capillaries in two awake mice, within the cortical depth of 0-200 µm. The statistical comparison in panel **a** was carried out using Student’s t-test. The statistical comparisons in panels **b-c** were carried out using ANOVA followed by a Tukey-HSD post-hoc test. The asterisk symbol indicates P<0.05.

Measurements of the RBC longitudinal size vs. capillary diameter, RBC speed, and line-density are presented Figure Author response image 1. The mean RBC longitudinal size in capillaries with smaller diameter (2-3 µm) was just slightly larger than in capillaries with diameters equal 4-6 µm, which may be expected due to more squeezing of the RBCs in the thinner capillaries, although the difference was not statistically significant in our measurements (Author response image 1). Here, the larger-diameter group included capillaries with diameters equal 4-6 µm instead of suggested 5-6 µm. This was done in order to increase the number of capillaries in the larger-diameter group. We observed a trend of increased mean RBC longitudinal size with the RBC speed (Author response image 1), where the RBC longitudinal size in the fastest group of capillaries (1-1.5 mm/s) was statistically significantly different than in the other two groups with the lower RBC speed. Finally, capillaries with the lower-line-density had more elongated RBC size than the capillaries with the median- and higher-line-density (Author response image 1; no statistically significant differences).

Finally, paired measurements of the RBC speed by the line-scan and point-scan method are presented in Author response image 2. Median RBC speeds measured by the line-scan and point-scan method were 0.61 mm/s and 0.69 mm/s, respectively. The difference between the RBC speed values obtained by two methods did not reach statistical significance.

**Author response image 2. respfig2:** Comparison between the RBC speed measurements by the line-scan and point-scan method. The measurements were performed in 58 capillaries across two mice within cortical depth of 0-200 µm. The statistical comparison wascarried out using Student’s t-test, but no statistical significance was found.

Altogether, the RBC longitudinal size varied both in the same capillary as well as between different capillaries as a function of capillary diameter, RBC line-density, and RBC speed. The differences between the mean RBC longitudinal sizes when measurements were grouped by capillary diameter, RBC line-density, and RBC speed were generally not large, reaching statistical significance only in the case of the high RBC speed group (Author response image 1), although with the limited sample size of our measurements. However, the fluctuation of the RBC longitudinal size over time in each capillary was moderate (STD = 2.3 ± 0.6 µm). Therefore, we believe that for the purpose of providing mean values and conducting group comparisons, our RBC-passage based RBC speed measurements, while limited by assuming the constant RBC longitudinal length, are still reasonably accurate, as evidenced by the small difference between the mean RBC speed obtained in paired measurements (Author response image 2). However, as Reviewer correctly pointed out, instantaneous RBC speeds obtained by the RBC passage method may have larger measurement error.

Since the major findings related to Figure 3 and Figure 4 are sufficiently supported by the RBC flux measurements alone, in the revised manuscript we present the RBC speed measurements from these figures in the Supplementary data, as we believe that they represent a valuable additional support to our findings and may also be of general interest to the research community interested in cortical capillary blood flow distributions.

Regarding the effect of the RBC speed measurements on the EATs estimation, we would like to clarify that RBC-PO2, InterRBC-PO2 and EATs were calculated without involving RBC speed (e.g., the results in Figure 6B,C and Figure 7C in the original manuscript). Instead, peaks and valleys of the phosphorescence intensity recordings (Figure 1C) were directly used such that RBC-PO2 was calculated with all the phosphorescence decays in the valleys (RBC-passages) in the segmented phosphorescence intensity time course (Figure 1C), InterRBC-PO2 was calculated with the decays in the central 40% of the peaks (plasma), and EATs were calculated as RBC-PO2 – InterRBC-PO2. In the revised manuscript, we better clarified this in subsection “Calculation of capillary PO2 gradients and EATs”, by stating that:

“RBC-PO2 was calculated with all the phosphorescence decays in the valleys (RBC-passages) in the segmented phosphorescence intensity time course (Figure 1C). InterRBC-PO2 was calculated with the decays in the central 40% of the peaks (plasma). EATs were calculated as RBC-PO2 – InterRBC-PO2.”

We agree with the reviewer that, because EATs were estimated based on average RBC-PO2 and InterRBC-PO2 from the passage of multiple RBCs (Lecoq et al., 2011), EATs estimation represents an average EATs value from the file of RBCs passing through the capillary segment during measurement.

Finally, Figure 6A in the original manuscript presented the only EATs-related data that was dependent on the RBC speed measurements. As suggested, we replaced it with the equivalent figure where PO2 was presented as a function of time from the RBC center (instead of distance that relies on instantaneous RBC speed measurement). Two versions of the Figure 6A are presented in Author response image 3 for comparison. The earlier version of Figure 6A is now presented in the Supplementary data, as we believe it presents a valuable complementary view on the PO2 distributions between RBCs. The potential effect of the instantaneous RBC speed measurement error on the presented data was discussed in the Supplementary text that accompanies the figure.

The following text was added to Discussion section to better clarify the use of RBC speed measurements by the RBC passage method:

“Besides, the RBC speed was calculated by assuming a constant RBC size along the capillary axis, without considering its potential variation due to RBC speed, hematocrit and capillary diameter. In a separate set of measurements (n = 2 awake mice), we performed line-scan measurements (Kleinfeld et al., 1998) in 58 capillaries (2-s-long measurements, 2 kHz line-scan rate) and obtained very close mean RBC speed values by processing the data using two methods: by estimating the angle of the RBC-shadow stripes (mean RBC speed = 0.61 mm/s) and by the RBC-passage technique used in this manuscript (mean RBC speed = 0.69 mm/s). The mean RBC longitudinal size was estimated by following the procedures described in (Chaigneau et al., 2003), and it did not vary significantly as a function of RBC speed, line-density and capillary diameter, except for the fast RBCs (>1 mm/s), which are also associated with the noisier measurements by both techniques. However, the fluctuation of the RBC longitudinal size over time in each capillary was moderate (STD = 2.3 ± 0.6 µm). Therefore, group comparison of the mean RBC speed values measured by the RBC-passage method may be performed with the reasonable accuracy, but instantaneous RBC speeds obtained by this method may have larger measurement errors. These limitations of the technique should be considered for particular experimental designs.”

**Author response image 3. respfig3:** 

Subsection “Calculation of capillary RBC flux, speed, and hematocrit”: The authors make the same mistake as Parpaleix et al., (2013) and Lyons et al., (2016) in the way they estimate hematocrit, which is normally a measure of blood volume percentage: "Hematocrit was estimated as the ratio of the combined duration of all valleys associated with the RBC passages to the duration of the entire time course." Because RBC elongation varies with velocity (see first comment), the hematocrit calculated in the paper will depend on velocity. The authors should name differently what is actually measured.It is difficult to understand how RBC PO2 was determined. Could the authors elaborate on their approach: did they consider the first bin ("micron") or an average several bins to determine RBC PO2? Additionally, the use of the criterion of "the central 40% of the peaks in the binary segmented time course" to extract InterRBC-PO2 could be prone to error. Given that the InterRBC-PO2 is defined at the lowest value of plasma PO2 reached between the passage of RBCs, when the RBC flux is low, the use of the central 40% criterion could yield an accurate value of the this parameter, but is likely to become less accurate with increasing RBC flux, as the period when the plasma PO2 is at its minimum will be shorter, and the inflection will be sharper. The use of this criterion should be validated, comparing the InterRBC-PO2 value it provides with those values that are extracted from more restricted windows at greater distance from the RBCs.All statistics are based on Student's t-test whereas in almost all comparisons, ANOVA test with multiple comparison analysis should be used.

A similar concern about ‘hematocrit’ calculation was raised in comment #1b by reviewer 3. As suggested by reviewer 3, in the revised manuscript, we replaced the word hematocrit with the RBC line-density, which more accurately reflect our measurements.

**Author response image 4. respfig4:** Illustration of the selection of phosphorescence decays for the calculation of InterRBC-PO_2_.

EATs were calculated. The selection of the phosphorescence decays for InterRBC-PO2 calculation is illustrated in Author response image 4. To investigate the effect of applied window on the calculated InterRBC-PO2 values, we recalculated the InterRBC-PO2 by using the central 40%, 30%, 20% and 10% of the phosphorescence decays in the peaks of the segmented phosphorescence intensity time course. The EATs were also recalculated by using the InterRBC-PO2 values based on different selection window sizes (e.g., 10%-40%). The results are shown in Author response table 1. As the reviewer speculated, the InterRBC-PO2 exhibits a trend of increasing with the enlargement of the selection window. However, differences between the InterRBC-PO2 (and also between EATs) for selection window sizes between 10% and 40% are very small (within 1 mmHg), which is generally below the measurement error and it doesn’t affect any conclusions involving EATs in the manuscript.

**Author response table 1. resptable1:** InterRBC-PO2 and EATs dependence on the size of the selection window. The measurements were acquired in 373 capillaries in n = 7 awake mice. Data are expressed as mean ± SEM.

Selection window size	InterRBC-PO2 (mmHg)	EATs (mmHg)
40%	38.0 ± 3.7	12.0 ± 1.7
30%	37.7 ± 3.7	12.2 ± 1.7
20%	37.4 ± 3.7	12.4 ± 1.8
10%	37.0 ± 3.6	12.7 ± 1.8

Our RBC-PO2 calculation is also slightly different from the protocol used in (Parpaleix et al. 2013). In (Parpaleix et al. 2013), the RBC-PO2 was calculated using the phosphorescence decays at the border of the RBCs within 4-ms-wide window, while in this work we used all the phosphorescence decays in the valleys of the segmented phosphorescence intensity time course. We recalculated all RBC-PO2 values by following the procedure outlined by (Parpaleix et al., 2013). This resulted in the average RBC-PO2 equal to 52.3 ± 4.0 mmHg, which is almost the same as the average RBC-PO2 reported in our manuscript (54.0 ± 4.0 mmHg).

In the revised manuscript we applied ANOVA test to all data where appropriate. For details, please see our response to comment #1.

Subsection “Calculation of capillary PO2 gradients and EATs” and subsection “Accumulated oxygen extraction fraction increases in deeper cortical layers”: The authors estimate layer-specific oxygen extraction fraction by comparing the PO2 in arterioles and venules at a given depth. This approach tacitly assumes that a layer-specific network of microvessels connects arterioles and venules in this layer, and that all RBCs which leave arterioles in the given FOV and layer flow into venules in that same field of view and layer. The authors should cite which theoretical and experimental papers support the hypothesis. In addition, as they have the tools to reconstruct easily the vascular angiograms, they should trace a series of pathways from the pre-capillary arteriole to the post-capillary venule in order to verify the hypothesis, which is key for the paper. They could potentially limit their investigation to layer I and IV pathways, which are likely to differ.

We agree that this is an important question. In some of our previous publications we have done the reconstruction of vascular angiograms (i.e. segmentation and graphing), but this procedure and the analysis needed to validate this hypothesis are not easy, and it may take long time to complete. However, in a recent theoretical paper by Schmid et al., (Schmid et al., 2017), it was shown based on large mouse brain microvascular angiograms acquired by David Kleinfeld’s group at UCSD that RBCs are predominantly moving in-plane and that “no significant movement in the direction of the cortical depth takes place.” In the same work, Schmid et al., found layer-specific differences in the flow and pressure distributions in the cortical vasculature. To better clarify this important point, we added the following text into Discussion section:

“We estimated the depth-dependent OEF, which implies existence of a laminar flow pattern in the brain cortical microvascular network. This was recently confirmed by Schmid et al. (Schmid et al., 2017), who applied numerical modeling of blood flow and tracking of the trajectories of individual RBCs in realistic mouse cortical vasculature, which led to conclusion that RBCs predominantly flow in plane and no significant RBC flow in the direction of cortical depth takes place.”

Is the mouse's head rotated around the rostro-caudal axis during the 2PLM sessions? If not (and unless the objective lens can be inclined, thought this doesn't appear to be the case in Supplementary Figure 1(b)), due to the inclination of the cortical surface at the coordinates of the barrel field it is probable that there is a discrepancy between the reported measurement depths and the true depth of measurement in the cortex. As illustration, the approximately 40-45 degree inclination of the cortex relative to the horizontal at the level of the S1 barrel field (as per Paxinos and Watson), would mean that the maximum reported imaging depth of 600 micros would in fact correspond to a true cortical depth of approximately 460 microns. Thus, the assignation of the measurements to different layers will be compromised. Conversely, if the mouse head is rotated, could the authors note this and comment on the likely effect on the comfort and behavioural state of the mouse during the imaging sessions.

In our experiments, the mouse head was rotated to make the cortical surface perpendicular to the optical axis. The center of the cranial window in our preparation is 2 mm posterior from Bregma and 3 mm lateral from the midline. We estimated that the rotation angle was ~35°. However, mice were habituated during training to such rotation and they exhibit normal and relaxed behavior during experiments. Therefore, we do not expect that head rotation had adverse effect on the results of our experiments. In the revised manuscript, we clarified this point in subsection “Animal preparation”, which now reads:

“The training was conducted while mice were resting on a suspended soft fabric bed in a home-built platform, under the microscope. Mice were gradually habituated to longer periods (from 10 minutes to 2 hours) of head-restraint with the head slightly rotated (~35°) to make the cortical surface perpendicular to the optical axis. All mice were rewarded with sweetened milk every 15 minutes during both training and experiments. While head-restrained, the mice were free to readjust their body position and from time to time displayed natural grooming behavior.”

Biological:3D projections of PO2 maps (Figure 1G) are made with acquisitions lasting only 0.6 seconds (2000 decays). This is really short as PO2 varies with time. It is clearly shown in Figure 5. PO2 may change by more than a factor 2 within 9 s. Therefore, PO2 maps using such brief acquisitions will change dramatically from one acquisition to the other. What is the scientific value of such 3D map? In addition, it artificially increases the number of vessels imaged.If this approach was also used to build Figure 2, it strongly decreases its significance.In the absence of proper statistics, Figure 3, Figure 4, Figure 5, Figure 6 and Figure 7 cannot be interpreted.In their 2014 paper, the authors reported that PO2 decreases with the capillary order. Could the author verify if their findings hold true in the awake mouse? This would require a simple analysis regrouping A1, A2 and A3 capillaries.Given the concerns raised above about the definition of RBC-PO2 in capillaries and the method for estimation of OEF by comparing arterioles and venules (and indeed A1-A3 and V1-V3 capillaires) in the same layer, it follows that the estimation of the relative contribution of different vascular compartments and the related conclusions (Subsection “Low oxygen extraction along the superficial capillary paths contributes to an increase in the mean venular SO2 towards cortical surface”) are questionable.

We do not anticipate that temporal fluctuations of PO2 and blood flow confound mean values of our measurements based on relatively short measurement intervals but in a large number of capillaries at rest. It is also not entirely clear why longer acquisitions are needed and how they may help. Longer acquisition intervals in each capillary would also force us to reduce number of sampled capillaries, which is a typical trade-off in in vivo measurements. Since the blood flow (and PO2) differ significantly between different capillary branching orders (likely more than the amplitude of temporal fluctuations in individual capillary segments), reducing the number of sampled capillaries while increasing the acquisition time per capillary may lead to less accurate estimates of the mean values from the capillary populations.

Figure 1G does not represent a 3D PO2-distribution ‘snapshot’ and we expanded the Figure 1 caption to better clarify this point. However, we believe it is still very valuable to present this PO2 distribution in the complex 3D microvascular network. With all the imperfections due to temporal fluctuations correctly pointed by the reviewer, the PO2 distribution in Figure 1G from arterioles to capillaries to venules is still very reasonable and we believe that it will be very useful to the readers to grasp the concept of cortical microvascular oxygenation. Please note that interpolated PO2 values were only used in Figure 1G.

In the revised manuscript we applied proper statistical analysis. Please see our response to comment #1.

In this study, capillary branching orders were manually identified in a limited number of segments per branching order, which prompted us to group the A1-A3 or V1-V3 capillaries together for analysis. Expanding the data set to respond to reviewer’s suggestion may require significant additional effort, but we believe that this is out of the scope of this manuscript.

Finally, we believe that by responding to the reviewer’s previous comments we alleviated concerns raised at the end of comment #6. For the details about RBC-PO2 calculation and validation, please see the response to comment #3. For the concerns about existence of the laminar blood flow in the cortex, please see the response to comment #4.

Reviewer #2:

The article presents novel data on hemodynamic states in the different layers of the whisker barrel cortex of awake mice. in vivo data are acquired using a novel oxygen probing technique with two-photon phosphorescence lifetime microscopy. Specifically, steady averaged oxygen pressures and red blood cell (RBC) flux counts were acquired. In addition, temporal variations within a 9s observation window were collected. A detailed statistical analysis of the layer dependence of oxygenation and RBC fluxes is presented. The main conclusions include the experimental observation of depth dependence of oxygen extraction, and a reduction in the RBC flux variability with increasing cortical depth. This study presents an invaluable experimental body of work that will help to elucidate oxygen extraction mechanisms in the mouse cortex and create in vivo data for mechanistic models to explain the physiochemical principles that drive and control cerebral blood flow and metabolism. This is a significant piece of work which I highly recommend for publication after modifications listed below:Technical comments:Introduction. The EAT is used to characterize the heterogeneity of the oxygen extraction within capillaries. What is the physical rationale for correlating oxygen point measurements to intracapillary resistance to oxygen delivery? Are these measurements for erythrocyte bound versus unbound oxygen in plasma? How is this EAT related to oxygen saturation?

Our EATs measurements were conducted with the goals to: (i) enable SO_2_ estimation in the capillaries, for which we needed RBC-PO2 measurements, (ii) explore the capabilities of the novel oxygen probe to provide faster and/or more accurate EATs measurements, and (iii) explore EATs dependence on the capillary branch order, RBC line density, cortical layer, etc. The particulate or discrete nature of RBC flow in capillaries causes EATs. Modelling studies (Hellums, 1977; Golub and Pittman, 2005; Barker et al.,. 2007) showed that larger difference between RBC-PO2 and InterRBC-PO2 (i.e., larger EATs) is associated with an increase in the intracapillary resistance to oxygen transport to tissue from capillaries. Due to non-zero EATs, mean capillary PO2 cannot be used in the Hill equation to compute capillary SO2. Instead, RBC-PO2 measured in the plasma in close proximity of the RBCs was used for that purpose, as an approximation of the mean PO2 inside the RBC.

It appears that measurements of RBC fluxes (RBC counts) assume that signals are generated from individual RBCs that are sharply separated. Figure 1C shows signals with different durations (and intensity). Is it possible that two aligned RBCs (two or more RBCs in file) may cause a longer signal that is indistinguishable from a single slow RBC? How would this affect the analysis of RBC fluxes?

The RBC-touching phenomenon may cause underestimation of RBC flux. In a subset of capillaries, we identified ~6% of valleys that appear to be ‘touching’ RBCs, indicated by relatively larger width and small spikes in the middle of the valley (please see an example valley denoted by an arrow in Author response image 5).

**Author response image 5. respfig5:** A representative 0.4-s-long recording of the RBC passages. The blue curve is the experimental time course, and the red curve is the fitted time course. The black arrow points to a small intensity ‘spike’, suggesting that 2 contiguous RBCs passed through the focal volume.

In the Introduction, the authors cite the next generation of biophysical models, which with the exception of Gagnon's 2016 paper do not correspond to the anatomical detail presented in this study. Layer specific oxygen consumption has been predicted by recent biophysical models that match the detailed anatomical scope and three-dimensional resolution of the proposed experimental study. For example, three dimensional predictions of depth dependent oxygen gradients in mouse are given in Gould et al., 2017. Biophysical models to predict depth dependent oxygen gradients in humans are presented in Linninger et al., 2013 and Gould and Linninger, 2015. These studies have been successful in solving oxygen delivery to brain tissue coupled with biphasic blood flow in a realistic cortical microanatomy and are therefore perfectly aligned with the scope of the current study. These advancements should therefore be incorporated and discussed.

We thank the reviewer for the suggestion. In the revised manuscript, we referred to the above manuscripts in Introduction and Discussion section.

Subsection “Accumulated oxygen extraction fraction increases in deeper cortical layers”. The data in Figure 2 seem to refer to steady magnitudes or levels, the use of the word "amplitudes" seems ambiguous.

We thank the reviewer for the suggestion. In the revised manuscript, we replaced the word ‘amplitudes’ in in subsection “Accumulated oxygen extraction fraction increases in deeper cortical layers” in original manuscript with the word ‘levels’.

Subsection “Capillary RBC flux and oxygenation homogenize in deeper cortical layers”. The prediction that homogenization of RBC flow velocity enhances the oxygen extraction is made based on a single segment analysis in the cited biophysical model. Here, experiments are presented for a three-dimensional vascular network. I am uncertain whether the two scopes are really "in line" with each other as suggested in the current text.

In the revised manuscript, we changed the text, which now reads:

“This result suggests that RBC flux in the deeper cortical layers is more homogeneous, which may facilitate oxygen extraction as theoretically predicted (Hartung et al., 2018; Jespersen and Østergaard, 2012; Schmid et al., 2017).”

Figure 5. The trend line in Figure 5 does not intersect at zero. What does the non-zero intercept of roughly 16-18mmHg pO2 mean? Should the trend have a zero intercept?

Non-zero intercept likely means that PO2 is not zero when RBC flux is zero, which is not physiologically impossible, as plasma may still be flowing even when the RBCs are not. In addition, capillary may be close to one or several more oxygenated vessels (arterioles or capillaries), creating the oxygen influx from the tissue into the capillary. In fact, in Figure 7—figure supplement 4, we reported a capillary having stalled RBC flow, and the PO2 was measured to be 15 mmHg, close to the intercept PO2 value.

Subsection “Capillary RBC flux and oxygenation homogenize in deeper cortical layers”. The number of segments in layer IV and V have only half the segments than layer I-III. Could this affect the statistics? If not, this observation should be listed in the limitations.

At larger cortical depths, expanding shadows below large pial vessels, partial obstruction of the optical paths by the edge of the cranial window, and gradual loss of resolution, contributed to lower number of reported measurements in the capillary segments. However, more than 100 capillary segments contributed to data analysis even from the deepest layer that we investigated and reported statistically significant differences between different cortical layers were obtained by applying proper statistical analysis. It is still possible that due to smaller number of measured capillaries in the deeper cortical layers, some investigated variables did not show statistically significant difference between cortical layers. We commented on this limitation in the Discussion section, which now reads:

“Another limitation is that at greater cortical depths, expanding shadows below large pial vessels, partial obstruction of the optical paths by the edge of the cranial window, and gradual loss of resolution, contributed to the smaller sampling size. It is possible that due to the smaller number of measured capillaries in the deeper cortical layers (e.g., 104 capillaries in layer V vs. 400 capillaries in layer I; Figure 3), some investigated variables did not show statistically significant difference between cortical layers.”

The assignment of upstream capillaries A1-A3 and downstream capillaries V1-V3 was done manually. Is it hard to imagine that roughly hundred segments (=according to subsection “Low oxygen extraction along the superficial capillary paths contributes to an increase in the mean venular SO2 towards cortical surface”, the analysis included 97 segments for all mice specimen) were identified by hand without the aid of image filters (e.g. Blinder et al., 2013; Hsu et al., 2017) in combination with automatic segmentation.

The identification of the branching orders in 97 capillaries in this manuscript was indeed performed manually by visual inspection.

In subsection “Low oxygen extraction along the superficial capillary paths contributes to an increase in the mean venular SO2 towards cortical surface” it is argued that the manual method provides statistical power. It is not clear what statistical power is referred to? I suspect that the results could come out quite differently, if labels were assigned differently. Is there any data on operator dependence of the labeling method? I recommend to consider writing the Discussion section more cautiously to reflect that results are based on an operator dependent method with high uncertainty in segmentation, and that major results would not be affected by uncertainty associated with operator dependence in upstream and downstream labeling. This point is optional to the discretion of the authors.

We apologize for the confusion. We wanted to say that grouping data from 3 consecutive branching orders (e.g., A1-A3 and V1-V3), instead of individual ones, helped with reducing the data variance in the presence of limited sample sizes for individual branching orders. In the revised manuscript, we rephrased the sentences in subsection “Low oxygen extraction along the superficial capillary paths contributes to an increase in the 251 mean venular SO2 towards cortical surface” in the original manuscript, which reads now:

“Due to a limited number of capillary segments with assigned branch order, A1-A3 and V1-V3 capillaries were grouped together to enable group comparisons with stronger statistical power.”

We verified the results in Figure 7 by randomly choosing 2 branching orders from A1-A3 and 2 branching orders from V1-V3, e.g. A1,2 vs. V1,2, A1,3 vs. V1,2, A1,3 vs. V2,3, etc. The final values were slightly different, but the main conclusions were the same. We discussed how the manual selection was done and the operator dependence. However, in light of current uncertainty how to define precapillary arterioles and where true capillaries start, automatic selection may not be more helpful. In the revised manuscript, we updated the relevant text in the Discussion section, which now reads:

“The capillaries were identified based on their morphology, without taking into account the smooth muscle cell coverage and pericyte types (Attwell et al., 2016; Hall et al., 2014; Hartmann et al., 2015; Hill et al., 2015; Mishra et al., 2014; Peppiatt et al., 2006; Secomb, 2017), and the assignment of capillary branching order indices was performed manually, which may be operator dependent. In principle, misclassifying vessel types and/or branching orders could potentially influence our analysis. However, based on the overwhelmingly larger number of capillaries compared to the non-penetrating arterioles and venules, our conclusions in general are unlikely to be different.”

Subsection “Low oxygen extraction along the superficial capillary paths contributes to an increase in the mean venular SO2 towards cortical surface”. The hematocrit changes were reported between upstream (A1-A3) and downstream capillaries (V1-V3). Why did the study not explore layer dependence or branch hierarchy dependence (Strahler order analysis) of hematocrit?

We did not observe statistically significant difference between RBC line-density values in the upstream (A1-A3) and downstream (V1-V3) capillaries, and between different cortical layers. Limited size of the sample with the labeled branch order prevented us from exploring the hematocrit dependence on the capillary branch hierarchy. In the revised manuscript, in subsection “Capillary RBC flux and PO2 are more homogenous in the deeper cortical layers”, we stated that:

“We did not find statistically significant difference in the values of the absolute RBC line-density, and its STD and CV between different cortical layers.”

Discussion section. The RBC flux is equal to the product of bulk flow rate and hematocrit (volume fraction). How should we understand the variance reduction in RBC flux without any relation to variations in hematocrit, since the quantities are directly related?

It is possible that changes in the RBC flux are balanced by similar changes in the RBC speed, as these two variables are highly positively correlated (for example, see Figure 7—figure supplement 3C and Figure 7—figure supplement 4C). In this case, relative hematocrit changes may be significantly smaller than the relative RBC flux and speed changes. We agree that it is plausible that some hematocrit changes may accompany the RBC flux changes, but were not resolved in our measurements.

Discussion section. The wide variability of hemodynamic states (high heterogeneity of capillary flow and oxygenation) was predicted previously in a biophysical model by Gould et al.,.2017. It would be helpful if the experimental results could be aligned with already completed theoretical work that aims at addressing the same points as those that are here so elegantly presented experimentally.

We thank the reviewer for the suggestion. In the revised manuscript, we expanded the Discussion section to better address findings in this and other suggested studies. For details, please see the response to comment #3.

In the same vein, it is worth mentioning that a main finding of the reduced variance of RBC fluxes was just recently predicted covering the whisker barrel mouse cortex with extension to the entire MCA territory (Hartung et al., 2018). The experiments confirm several findings so that the predictive model results are highly relevant for this study and should be discussed.

Yes, we agree, and we are thankful to the reviewer for pointing us to this reference. The new text was added in the Discussion section, which states:

“Layer-dependent homogenization of blood flow has been predicted by modelling the blood flow distribution in large-scale mouse brain microvasculature covering the whisker barrel cortex (Hartung et al., 2018).”

Reviewer #3:

The work addresses highly relevant questions (depth-dependent difference, homogeneity/heterogeneity of microvascular flow and oxygenation). The amount of measurements performed is significant and the capabilities of the new oxygen probe PtTAPIP to measure a variety of blood flow characteristics in large cortical depths is nicely demonstrated. However, we have major concerns regarding some of the analysis performed and some of the conclusions drawn (listed below). Furthermore, there is a significant shortcoming in referencing earlier work. It remains unclear, where the current work goes beyond what was published before.Essential revisions:1) Calculation of capillary RBC flux, speed and hematocrit (subsection “Calculation of capillary RBC flux, speed, and hematocrit”):As the authors are well aware, the methods presented to compute RBC flux, RBC speed and hematocrit are only valid in vessels where single file flow persists. The authors write that those measurements were performed in "capillaries". However, it remains unclear how capillaries were identified as capillaries. Were diameter measurements performed? As the RBC diameter of the mouse is 6 µm, single file flow can only be expected for vessel diameters < 6 µm. This aspect is key for the validity of the results.We propose that the authors come up with a table describing the measured vessels (type, depth, number) for clarity.

The capillary size permitting single-file flow may not be limited to <6 µm, due to the glycocalyx thickness and, possibly, an additional plasma layer between RBCs and glycocalyx ((Kim et al., 2009; Fedosov et al., 2010), SfN 2018 poster presentation 318.14/VV10). Variability of reported capillary diameters is relatively high, while some measurements estimate even the mean cortical capillary diameter of a mouse above 7 µm (Cai et al., 2018). Such variability in measurements, to some extent, may be attributed to a problem of defining a capillary (Hall et al., 2014; Mishra et al., 2014; Hartmann et al., 2015; Hill et al., 2015; Attwell et al., 2016; Secomb, 2017). Selection of capillaries based on identification of smooth muscle cells and pericytes (and their subtypes) was not performed in our study due to significant complexity that such procedure will add, especially considering a large sample size used in our analysis. Importantly, it may not provide more accurate results, as the debate on what is a proper classification of vessel-types is still ongoing. Instead, we relied on vascular morphology as a guide. Capillaries were typically identified starting one or two segments away from the diving arterioles and surfacing venules based on their morphology (i.e. smaller diameters and higher tortuosity). Importantly, the deep modulation of the phosphorescence intensity time courses indicate that RBCs are flowing in a single-file in capillaries we selected for measurement of PO2 and RBC flux (e.g., Figure 1C). The segmentation of the phosphorescence intensity time courses was evaluated by the coefficient of determination (R^2^) between the experimental and fitted time course, and the data with R^2^<0.5 was rejected according to the previously established protocol (Lee et al. 2013). We updated the Methods section to better describe selection of capillaries and processing of the phosphorescence intensity time courses.

In the legend of each figure, we provided the information about animal number, sample size, and imaging depth. By following the reviewer’s suggestion, we also include a Supplementary Table with the detailed information in one place about sample sizes, vessel types, imaging depths, and animal numbers (Author response table 2).

**Author response table 2. resptable2:** Measurement information for the main analysis in Figures 2-7.

**Parameters**	**Depth**	**Numbers of samples/mice**
**Capillaries**	**Arterioles**	**Venules**
**Mean-PO_2_**	0-600 µm	978/15	11/7	14/7
**RBC Flux**	0-600 µm	978/15	N.A.	N.A.
**Temporal Fluctuation**	0-600 µm	373/7	N.A.	N.A.
**EATs**	0-600 µm	373/7	N.A.	N.A.
**Branching Orders**	0-300 µm	97/5	N.A.	N.A.

1a) Additional Concern regarding velocity calculations (subsection “Calculation of capillary RBC flux, speed, and hematocrit”):Characteristics Mouse RBC: Diameter 6 µm, Volume: 45.5 µm3 (Windberger et al., 2013), Thickness: 1.6 µm.Assuming that each RBC is 6 µm is not correct. Especially, in larger vessels the RBC orientation makes a significant difference in the proposed velocity calculation (by a factor of ~4).For vessel diameters <5.5 µm the applied velocity calculation is more suitable, because here RBCs need to squeeze into the vessel. However, also here we need to account for changes in RBC length depending on the vessel diameter.In summary, the presented approach gives at best a rough estimate of the RBC velocity and it remains unclear, why the authors have not chosen line scans that would be so much more appropriate.

This is a very good point raised by the reviewer. We addressed this concern in response to comment #2 by reviewer 1.

1b) Additional Concern regarding hematocrit calculations (subsection “Calculation of capillary RBC flux, speed, and hematocrit”):The hematocrit computation is based on the assumption that plasma and RBCs have the same velocity. However, RBCs travel on average faster than plasma (Fahraeus effect). This velocity difference can cause an underestimation of hematocrit. Even so this approach is used frequently this assumption should be described and the term "line-density" should be used instead of "hematocrit".

We agree and thank the reviewer for the suggestion. In the revised manuscript, we replaced word ‘hematocrit’ with ‘RBC line-density’.

1c) Additional Concern regarding flux and velocity calculation (subsection “Calculation of capillary RBC flux, speed, and hematocrit”):For the flux and velocity computation it is crucial that individual RBCs can be distinguished from another. Can this be guaranteed for all hematocrit levels? Some discussion/comments should be added.

A similar comment was raised by reviewer 2 (comment #2). For details, please see our response to that comment.

1d) Selection of vessels for analysis (Subsection “Capillary RBC flux and oxygenation homogenize in deeper cortical layers”, subsection “Imaging of EATs and capillary RBC flow”):The selection criterion for the vessels that have been chosen to analyse flow properties are not described. Moreover, it remains unclear why the number of vessels investigated per layer differs so significantly (400, 356, 118, 104).

The flow properties (e.g., RBC flux, speed and line-density) were measured only in capillaries. Please see our response to comment #1 for details about capillary identification.

At larger cortical depths, expanding shadows below large pial vessels, partial obstruction of the optical paths by the edge of the cranial window, and gradual loss of resolution, contributed to lower number of reported measurements in the capillary segments. Please see our response to related comment #7 by reviewer 2.

2) Quantification of the temporal fluctuations of capillary Mean-PO2, RBC flux, speed and hematocrit (subsection “Quantification of the temporal fluctuations of capillary Mean-PO2, RBC flux, speed, and hematocrit”, Discussion section):On the one hand, the chosen measurement time is rather short for a good averaging of the readouts. On the other hand, an averaging interval of 0.6 s is too large to resolve fluctuations in the capillary bed. An RBC with an average velocity of 1mm/s would travel 600 µm during that time. So, for the overall statistics, longer periods should be measured, but for describing the fluctuations over time, those segments should be split up into smaller than 0.6 s intervals, so that the fluctuations are dampened too much due to the averaging.The arguments given in the discussion for the chosen interval seem to be motivated by dynamics on a larger scale but not by microvascular fluctuations. Why is the rate of oxygen consumption relevant for the capturing microvascular flow dynamics?Alternatively, it has to be described more clearly that only fluctuations on the time scale of seconds are analysed and that faster fluctuations persist but are not of interest in the current study.

It is correct that both faster and slower fluctuations are present in the microvascular network. For example, resting state connectivity studies typically probe frequencies below 0.1 Hz, and respiration and heartbeat introduce frequency components that were smoothed out by our measurements. The temporal resolution and recording interval of our measurements are therefore limited to capturing fluctuations at a rate of oxygen consumption, as described in the original submission, and it is also sufficient for resolving the blood flow transient responses to short neuronal activation (typically within a few seconds for a short stimulus). In the revised manuscript, we expanded the Discussion section to better explain the scope of measurements of temporal fluctuations:

“The temporal resolution of our measurements (0.6 s) ensured that the blood flow and PO2 fluctuations due to cardiac and respiratory cycles, as they occurred in awake mice, were averaged out, but it is sufficient to capture the dynamics related to the rate of oxygen consumption, which could be estimated as the time for tissue PO2 to drop to zero after blood flow stoppage. As reported in cats, that time was at least several seconds (Acker and Lübbers, 1977; Whalen and Nair, 1975). In addition, blood flow responses to neuronal activation could typically be resolved with such temporal resolution (Uhlirova et al., 2016). However, it is important to note that our experiments probed just one frequency window within a wide range of both faster and slower fluctuations present in the microvascular network.”

3) Analysis of EATs:Intracapillary resistance of oxygen transport to tissue decreases in deeper cortical layers (subsection “Intracapillary resistance of oxygen transport to tissue decreases in deeper cortical layers”) and Low oxygen extraction along the superficial capillary paths contributes to an increase in the mean venular SO2 towards cortical surface (subsection “Low oxygen extraction along the superficial capillary paths contributes to an increase in the mean venular SO2 towards cortical surface”):In order to understand the observed EAT trends, it is crucial to also analyse RBC-PO2 and interRBC-PO2. This aspect has been neglected for most EAT results. As such, possible reasons for the observed EAT drop in Figure 6C and the EAT increase in V1-V3 segments have not been analysed sufficiently (Based on "Supplementary Figure 3a" RBC-PO2 does not drop? Thus interRBC-PO2 should rise for deeper layers to explain the EAT drop? How can a rise in RBC-PO2 and interRBC-PO2 for deeper cortical layers be explained? A higher interRBC-PO2 would however suggest a "higher intercapillary resistance to oxygen transport" instead of a lower one.)Moreover, the EAT strongly depends on the distance of the capillary to the arteriole/venule (see Figure 7C). This impact should be discussed, e.g. how do you make sure that your depth-dependent EAT average is not affected by a larger number capillaries close to arterioles?In an earlier work by the same authors, they did not observe EATs. The authors should discuss in detail where this discrepancy originates from.

The increase in interRBC-PO2 may not necessarily mean higher intravascular resistance (IVR) to oxygen transport to tissue. For example, increase in RBC line-density reduces IVR (Federspiel and Popel, 1986), but it is associated with an increased interRBC-PO2 (Lyons et al., 2016). The EATs may be affected by multiple parameters, such as RBC spacing, shape, and wall-to-RBC spatial clearance (Popel, 1989; Golub and Pittman, 2005; Lücker et al., 2017). In addition, our measurements provide average values from multiple RBC passages, not accounting for fluctuations of parameters influencing EATs, which may differ between cortical layers and/or proximal and distal capillaries and differentially affect the EATs measurements. The EATs did show a trend of dependency on the branching order (Figure 7C in the original manuscript), but the difference between EATs (as well as the RBC line-density) in proximal and distal capillaries did not reach statistical significance in our measurements. Altogether, this leaves a number of possibilities to address in order to understand the detailed mechanisms behind EATs changes across cortical layers and capillary branches. We believe that such work goes beyond the scope of this manuscript, but we are very interested to pursue it in the future studies. We added the following sentences at the end of the paragraph about EATs findings in the Discussion section to better address these points:

“However, it is important to note that mechanisms that govern the EATs values are multifactorial. The EATs may be affected by multiple parameters, such as RBC spacing, shape, and wall-to-RBC spatial clearance (Popel, 1989; Golub and Pittman, 2005; Lücker et al., 2017). In addition, our EATs measurements provide average values based on multiple RBC passages. Therefore, they do not account for fluctuations of parameters that are affecting EATs, which may differ between cortical layers and/or proximal and distal capillaries, and differentially affect the EATs measurements.”

Regarding the previous measurements of EATs by us and other groups, please see our response to comment #1 By reviewer 1.

4) Depth-dependent oxygen extraction (subsection “Accumulated oxygen extraction fraction increases in deeper cortical layers”, Discussion section):We generally agree with the idea to use the difference in oxygen saturation as an indicator of oxygen extraction. However, as also shown in this work here, many factors have an impact on the oxygen saturation. Thus, saturation difference is not always equal to oxygen extraction. For example, higher blood flow in layer I leads to a higher oxygen availability in layer I and thus to higher SO2 values in the venules, even if the oxygen extraction is constant over depth."Benefitting from the improved sensitivity of the new oxygen probe, we were able to measure intracapillary longitudinal PO2 gradients in a larger number of capillaries…" why does the application of the improved probe help to increase the number of vessels measured?

In this study, we did not directly measure total blood flow in different cortical layers. Without knowing the blood flow, as reviewer suggested, it is difficult to tell how much oxygen was extracted. It is also worth pointing that hypothetical case that reviewer created to illustrate his point, namely that blood flow is higher in layer I than, presumably, in layer IV, is probably opposite in the mouse whisker barrel cortex. While our experimental results do indicate that capillary RBC flux is somewhat (~13%) smaller in layer IV than in layer I (Figure 3A), measurements by us and others reported significantly higher capillary density in layer IV than in layer I (~83% higher based on Suppl. Figure 7b in Sakadzic et al., 2014; >50% higher based on Figure 2C, Gould et al., 2017). Combined, these measurements suggest that macroscopic blood flow in layer IV is higher than in layer I, which may be expected as other indicators imply higher oxygen metabolic rate in layer IV. Then, higher blood flow and OEF in layer IV than in superficial cortical layers suggest that oxygen extraction may be much larger in layer IV than closer to the cortical surface. We updated the text in the Discussion section to better clarify these points, which now reads:

“A faster decrease in SO2 with cortical depth in the ascending venules than in the penetrating arterioles resulted in a higher depth-dependent OEF in the deeper cortical layers, reaching the maximum in layer IV (Figure 2B). In addition, the total blood flow may be higher in layer IV than in layer I, as suggested by the measurements of capillary RBC flux (~13% lower in layer IV than in layer I; Figure 3A) and capillary density (>50% higher in layer IV than in layer I; (Sakadzic et al., 2014; Gould et al., 2017)). Altogether, this implies that oxygen extraction was higher in deeper cortical layers, which would be in agreement with the finding that layer IV has the highest neuronal and capillary density in mouse cortex (Blinder et al., 2013; Lefort et al., 2009; Patel, 1983; Wu et al., 2016), and that the cells in layer IV exhibit the highest cytochrome oxidase labeling activity, suggestive of the highest oxidative metabolism (Land and Simons, 1985).”

Comparing to the old version of the probe, Oxyphor2P (Esipova et al., 2019) has more red-shifted 2-photon excitation and emission, higher quantum yield, larger 2-photon absorption cross-section (overall, it is ~100X brighter probe), and more single-exponential decay. These improved properties enabled deeper imaging, reduced rejection of data with poor SNR, faster selection of capillaries due to improved contrast of survey scan images, and shorter acquisition times (we collected 30,000 phosphorescence decays per capillary to extract EATs, comparing to the 40,000-60,000 phosphorescence decays per capillary using PtP-C343).

[Editors’ note: the author responses to the re-review follow.]

The reviewers have discussed the reviews with one another and the Reviewing editor has drafted this decision to help you prepare a revised submission. All reviewers believe that the work is interesting. They however raised a number of questions that I ask you to address and which would clarify the manuscript. No new data is required. I have summarized the key questions:1) The demonstration that RBC size can be estimated from the 2PLM measurements is still not convincing. The way you analyzed your new data is not informative: Comparing the mean RBC values from 58 capillaries measured with Line-Scan and Point-Scan (Author response image 1) does not solve the problem, in particular as the variability is large. Please show the plot in Author response image 2, with lines joining each values, capillary per capillary. It is important as Author response image 1 already seem to demonstrate that RBC longitudinal size varies a lot. The 3 reviewers believe that if your new analysis reveals that RBC size cannot be accurately estimated, all RBC velocity data should be removed from the paper and the resulting uncertainty of RBC PO2 measurements be carefully addressed in the discussion.

In our previous response, we presented Author response image 2 (Comparison between the RBC speed measurements by the line-scan and point-scan methods), where the mean values were reasonably close, but the instantaneous speeds differed sometimes significantly, and the interquartile ranges were large. Therefore, our conclusion was that for the purpose of conducting group comparisons with the mean values, our RBC-passage based RBC speed measurements are still reasonably accurate. However, in agreement with reviewers, we concluded that instantaneous RBC speeds obtained by the RBC-passage method may have larger measurement error. While this conclusion about the instantaneous RBC speed measurements was reasonable and based on the differences between the simultaneously measured RBC speeds by the two methods, we agree that only the box-plot presentation of the data may not be the most appropriate in this case since large interquartile ranges may possibly result from perfectly pairwise-matched measurements, which was not the case here. As suggested by reviewers, we now provide both pairwise connections between RBC speed data points obtained by the two methods and the corresponding box plots (Author response image 6). We hope that this figure can help with clarifying the issue. Some additional results are also provided in the response to the comment #2 raised by the reviewer #1.

**Author response image 6. respfig6:** Comparison between the RBC speed measurements by the line-scan and point-scan method. The individual RBC speed values estimated by two methods from each capillary (green circles) are connected by green lines. Boxplots of the line-scan and point-scan RBC-speed values indicate the median values, 1^st^ and 3^rd^ quartiles, and maximum and minimum values. No statistically significant difference was found between the mean values (Student’s t-test). The regression slope between the paired RBC-speed measurements is 0.87 (not shown). The measurements were performed in 58 capillaries across two mice within the cortical depth range of 0-200 μm.

Based on the above results, we still think that for the purpose of providing mean values and conducting group comparisons, our point-scan RBC-speed measurements are reasonably accurate. We agree with reviewers, as we did in the previous response, that instantaneous RBC speed measurements may have larger measurement errors. Pairwise comparisons (Author response image 6) demonstrate this variability, which seems to be larger for larger RBC speeds (please note that line-scan method also has reduced accuracy at larger RBC speeds). Our RBC speed results may be of general interest to the research community interested in cortical capillary blood flow distributions and we would like to keep them in the Supplementary data. In the revised manuscript, the original Figure 7E that presents the mean RBC speed values in the upstream and downstream capillary branches was moved into the figure supplement. The limitation of the RBC speed estimation was previously explained in the Discussion section and, to further clarify it, we added the following sentence in the Results section:

“Here, please note that the RBC speed calculation was model-based by assuming a constant RBC size (6 µm) (Unekawa et al., 2010).”

Regarding the ‘resulting uncertainty of the RBC-PO2 measurements,’ we would like to clarify that, in this study, RBC-PO2 was calculated using the phosphorescence decays corresponding to the valleys (RBC-passages) in the phosphorescence intensity time course. The measured temporal width of the valleys is dictated by the instantaneous size of the RBC passing through the optical focus. The temporal width was estimated by directly fitting for a simple binary pattern, and it doesn’t involve assumption of the constant RBC size used in calculation of the RBC speed.The only EAT-related PO2 result that involved RBC-speed estimation is the distance-resolved PO2 gradients (Figure 6—figure supplement 1). Please note that the PO2 gradients and assumption about constant RBC size were not used to calculate any other PO2-related properties (e.g., Mean-PO2, RBC-PO2, InterRBC-PO2 and EAT).

Some uncertainty of the RBC-PO2 estimation may come from the fact that within each valley in the phosphorescence intensity time course, plasma PO2 likely continuously decreases as a function of distance from the RBC center (valley center). Due to spatial dimensions of the RBC and the excitation beam, when the excitation beam overlaps with the RBC center, measured PO2 likely best represents the PO2 in the space between the RBC and the capillary wall. As the RBC center moves away from the beam focus, the excited volume fraction of plasma region that is more distant from the RBC surface continually increases (assuming very simplistic ellipsoid shape of the RBC). In addition, the thickness of the plasma layer between the RBC and the capillary wall may be slightly different in capillaries with different diameter and for different RBC speeds, which may result in slightly different PO2 in this plasma layer. While the PO2 at the center of the valley likely best represents the PO2 at the surface of the RBC, measurement of PO2 when using the current imaging tools will be noisier if we consider only a very narrow central valley range.

As we previously described, we averaged the decays in the entire valley to calculate RBC-PO2. However, the PO2 change from the center to the edge of the valley may not be large, as evidenced by a PO2 plateau in the vicinity of the RBC (Figure 6—figure supplement 1). Importantly, prompted by the previous reviewer 1’s comment, we compared the RBC-PO2 calculations using the phosphorescence decays from the entire valley (mean RBC-PO2 calculated from 373 capillaries across n = 7 mice is 54.0 ± 4.0 mmHg (mean ± SEM)) and by following the protocol used in (Parpaleix et al. 2013), where PO2 was calculated using the decays at the border of the valley (mean RBC-PO2 = 52.3 ± 4.0 mmHg). The observed small PO2 difference is not statistically significant, it is inconsequential for the results presented in this manuscript, and further confirms that PO2 change across the valley is not large. Therefore, both approaches (averaging across the entire valley or close to its edge) may provide valid results.

We further clarified these points in the Discussion section by adding the following text:

“We averaged all the phosphorescence decays within the valleys (i.e., RBC-passages) in the segmented phosphorescence intensity time courses to calculate RBC-PO2. This approach does not consider that within the valley PO2 may vary as a function of distance to the center of the valley, and the extent of this variation may be different among capillaries of different diameter and/or RBC speeds.”

2) Please address in the discussion the issue that a saturation difference may depend on several factors.

We expanded the discussion about the oxygen extraction change across the cortical layers by adding the following sentence:

“Since oxygen extraction depends on both total blood flow and arterio-venous oxygen saturation difference, it will be important in the future to further experimentally investigate the total blood flow differences across different cortical layers.”

Please also see our answer to the comment #4 raised by reviewer 4.

3) Add the plot for interRBC-PO2 over depth to the supplementary figures.

We thank the reviewers for this suggestion. In the revised manuscript, plots for InterRBC-PO2 were added, and presented in Figure 3—figure supplement 1.

4) Add in the Impact Statement that the measurements are done at the steady state.

We thank the reviewers for this suggestion. In the revised version, the impact statement was updated, and it now reads:

“Resting-state capillary blood flow and oxygenation are more homogeneous in the deeper cortical layers, underpinning an important mechanism by which the microvascular network adapts to an increased local oxidative metabolism.”

The reviewers have added some comments (see the full reviews attached below) to which you could briefly answer.

Reviewer #1:

The authors have made new experiments, new analysis and the text has improved. Previous works are now better described and discussed. The statistics are more appropriate.The new experiments and analysis done in response to comments #2 and #3 are interesting but they do not fully address the issues raised. This concerns primarily RBC velocity measurements. The good news is that it could be easily done with simple analysis without any further experiment.The authors initially used the assumption of an identical average size of RBCs to base other calculations, in particular RBC speed. They now show new experiments which results are summarized in several "R1" plots. The plots show that RBC longitudinal size is very variable, increasing very significantly with RBC speed and spreading from 3 to 14 µm (R1C1261 c plot (<30% RBC line-density)). This clearly indicates that RBC shadows measured during PO2 acquisitions cannot be used to estimate RBC speed. Note that the authors should give the "n" for each of the groups of capillaries in all the new "R1" plots.This variability of RBC size is unfortunately masked in the R1C2-2 plot. This plot needs to show paired measurements in specific capillaries, the question being whether or not similar values of RBC velocity are recorded by the two methods when (presumably) similar real RBC velocities are measured. Showing population averages for the 58 capillaries assessed occludes the similarity/difference on a capillary by capillary basis. The important point here is that all the data are already acquired, for capillaries of known diameters. So the authors could now easily compare RBC values for paired measurements (with the 2 methods) in capillaries of different diameters. Please indicate the "n" for each of the groups of capillaries and plot, for each group, all RBC speed values measured with the 2 methods. I suspect that the authors will find differences with the two techniques. This would invalidate all RBC measurements obtained with point scan PO2 acquisitions.It is important that the authors clarify the point as fluctuations of RBC shadows, whether measured in distance or time from the center of the shadow, will modify the RBC PO2 value. In fact, it will decrease RBC PO2 and thus the EAT amplitude. Could the authors measure the decays as a function of time, but after an alignment at the RBC border? This would solve the problem.

We agree that only box-plot representation of the data in the previous response may not be sufficient to assess pairwise differences in the RBC speed measurements by the two methods. Please see our response to the summary comment #1 for additional details and the updated plot that includes pairwise connections between measurements. We believe that the previous written response addressed this limitation and hope that updated plot will provide additional clarification.

Regarding the extreme values of estimated RBC longitudinal sizes, please note that they were obtained from the fitted widths of the shadows and the RBC speeds, both of which were extracted from the line-scan images. Measurements of both these parameters have limitations. The line-scan method is less accurate for high RBC speeds, while RBC shadows sometimes may be due to stacked RBCs (as we pointed out previously) and they were noisier when acquired with the line-scan method (due to shorter dwelling time per time-point) than with the point-scan method. This is still not negating that there is an increased error of the instantaneous RBC speed measurements by the point-scan method. However, we contend the mean RBC speed values from the population of capillaries obtained by the two methods are reasonably close. We hope our explanation of this limitation in the manuscript is clear so that readers can be well aware of it while still having a benefit of assessing these measurements in the Supplementary data.

As reviewer requested, in Author response image 7 we compare the RBC speed values estimated by the two methods. The comparison was conducted separately with two groups of capillaries based on capillary diameter. The data scatter is notable, but the regression slopes in both groups are reasonably close to 1.

**Author response image 7. respfig7:** Correlation between the line-scan and point-scan RBC-speed values in the capillaries having the diameter of 2-3 µm (left panel) and 3-5 µm (right panel).

Regarding the RBC-PO2 measurements, we believe that averaging the phosphorescence decays in the entire valley in the segmented phosphorescence intensity time course is likely going to provide slightly larger RBC-PO2 values than with the decays only close to the valley edge (please see our answer to the summary comment #1), and that both measurement approaches are likely affected by the RBC shape (when RBC elongates at high speed, plasma layer between the RBC and the capillary wall becomes thicker and mean PO2 in this plasma layer likely slightly decreases). Therefore, it is not clear to us that measuring the decays as a function of time after aligning with the RBC border will solve the problem. However, the difference between RBC-PO2 values obtained by the two approaches are rather small comparing to the RBC-PO2 (please see our answer to the summary comment #1) and it looks to us that they both provide valid estimates of PO2 in the plasma adjacent to the RBC.

Reviewer #4:

[…] The additional studies are an important step to judge the accuracy of the RBC velocity measurements.However, the following major concerns remain:- Author response image 2 scatter plot that directly compares the RBC velocity from the line-scan and the point measurement would be more appropriate. Additionally, the average relative difference between the two measurements should be provided.- Author response image 7: The results clearly show that there is significant variability in the longitudinal RBC size (CV = 0.4, longitudinal sizes ranging from 4-14 µm). Moreover, the RBC longitudinal RBC size is correlated with the RBC speed and the RBC Line-Density. It is impossible to estimate the impact of these dependencies on the presented RBC speed results.

Please see our answers to the summary comment #1 and the comment #2 from reviewer 1.

I thank the authors for the additional explanations regarding IVR and I believe that the changes in the discussion are very valuable. However, some aspects of my initial comment have not been addressed.The EAT is computed from the RBC-PO2 and the interRBC-PO2. As such I believe that EAT, RBC-PO2 and interRBC-PO2 should always be analysed and discussed hand in hand. Thus, it would be valuable for the manuscript to add the plot for the interRBC-PO2 over depth to the supplementary figures and to discuss RBC-PO2 and interRBC-PO2. Moreover, I suggest to add and discuss the EAT STD & CV plots over depth as it is done for all other quantities.As stated in my original comment it is a surprising results that RBC-PO2 increases over depth and I don't know which mechanism could explain this increases (Figure 3—figure supplement 1). The same holds for interRBC-PO2 (which has to increase more than RBC PO2 in order to explain the EAT drop over depth). I do not ask for additional experiments, but I believe that is important to discuss these trends.In my initial question I asked how the author's ensured that the depth-dependent differences are not affect by the position of the chosen capillaries along the capillary pathway or to put it differently how the capillaries were chosen over depth to guarantee an equal distribution of "upstream" and "downstream" capillaries. I kindly ask the author to describe if this has been considered in some way? If it has not been considered the possible impact on the depth dependent results should be discussed.The EAT drop over depth is one of the major results of this manuscript and as the authors state in subsection "EAT measurements are typically much noisier"Discussion section. Consequently, I believe that the available data should be analysed as rigorously as possible.

In the revised manuscript, we included the InterRBC-PO2 plot in Figure 3—figure supplement 1, and we also included the STD and CV of EATs in Figure 6 —figure supplement 1. Smaller difference between RBC-PO2 and InterRBC-PO2 and, consequently, smaller EATs at greater depths are likely a consequence of more narrowly distributed capillary blood flow and oxygenation. It remains unclear to us how EAT STD and CV are related to oxygen delivery, and as a result, we are hesitant to include extensive comments on these parameters in the manuscript. Nevertheless, we are happy to provide these results as figure supplements.

We can only hypothesize as to why the mean capillary RBC-PO2 and InterRBC-PO2 increase with depth was observed. As reviewer mentioned, change in proportion of upstream to downstream capillaries with depth may contribute. The observed increase in mean values may be in part due to more homogeneous flow and PO2, since in the networks in which vascular oxygen content and blood flow are strongly positively correlated (which is the case here), mean vascular segment PO2 may be higher when PO2 and blood flow distribution is more homogeneous even for the same total blood flow and total oxygen flux through the tissue volume. Other mechanisms may also contribute, but we are not sure that discussion based on our available data will be insightful.

Regarding the proper sampling of the upstream and downstream capillaries, we believe that our measurement protocol (i.e., measuring PO2 in all capillaries identified within the field of view at each imaging plane) together with the large number of capillaries interrogated at each depth, ensured that upstream and downstream capillaries were sampled equally proportional to their natural number densities. Consequently, by following our measurement protocol, we could not guarantee an equal distribution of interrogated upstream and downstream capillaries unless the equal number of upstream and downstream capillaries is already naturally occurring across the cortical layers. However, regardless of the potential depth-dependent differences in the densities of upstream and downstream capillaries, we believe that our approach (i.e., sampling a large fraction of segments without discriminating them by branching order) is appropriate for obtaining the mean values at different depths. Please see our response to comment #1d from reviewer 4 and the updated subsection “Measurement of RBC-PO2, InterRBC-PO2, EAT, and RBC flow in capillaries” for clarification of the sampling protocol. Please also note that the ratio of the number of upstream to downstream capillaries may contribute to the mean value across all the capillaries at one depth, and that this ratio may be changing with depth.

I disagree with the given explanations why RBC flux in layer IV is supposed to be higher than in layer I. What matters here is not the average RBC flux and the higher capillary density per layer but the flux into the capillary bed per layer or to put it differently the flux out of the diving arterioles per layer.The given arguments connecting average RBC flux and capillary density are not plausible. I try to explain this with a simplified example. Imaging two similar tissue volumes: One with a single vessel and flow rate q1 through that vessel. The second one also has an inflow rate of q1 but the vessel splits in two vessels. In both cases the inflow (and thus the oxygen availability) per tissue volume is the same. The vessel density is however higher in the second one. The authors now argue that the macroscopic flow rate would be larger in the second example, which is not true.Of course, higher vessel density, i.e. more flow pathways, might have an effect on the overall flow rate. However, many open questions remain regarding these issues and simply relating RBC flux and capillary density to estimate the overall flow is not correct.The referenced figures (Suppl. Figure 7b in Sakadazic et al., 2014 and Figure 2c in Gould et al., 2017) show the number of capillary segments, which is not the same as vessel density.In the original work from Blinder et al., 2013 Figure 2c the capillary density increased from ~4% in layer I to ~5% in layer IV, which is an increase by ~20% but not by 50%.Taken together, my initial question remains, i.e. the depth-dependent blood flow/oxygen availability has a strong impact on the actual oxygen extraction per layer. This should be discussed properly. Maybe it would good to change the variable name to depth-dependent saturation difference or comparable, because the term oxygen extraction seems to be misleading.

We agree that what matters is the total blood flow that supplies the cortical layer. We also agree with the conceptual example that reviewer presented, but we believe that flow parameters used by reviewer in this example do not agree with our measurements. To better clarify it, let’s consider the same example with two similar tissue volumes: One with a single vessel and the flow rate q1, and the second one in which the vessel splits in two vessels but each of these two vessels has the flow rate q1 identical as in the single vessel in the first tissue volume. In this case the inflow (and thus the oxygen availability) per tissue volume is two-fold higher in the second sample. We believe that this example more closely represents the blood flow and capillary density in cortical layers I and IV. We measured only a small to moderate decrease (~13%) of the mean capillary RBC flux from layer I to layer IV, but capillary segment density in layer IV is much higher than in layer I (~50%, as measured by us and others). We believe that this supports our statement made in the manuscript that blood flow may be higher in layer IV than in layer I.

Regarding the measurements of the capillary densities in layers I and IV, probability density of capillary segments as a function of depth was presented in Figure 2C in (Gould et al., 2017) and number of capillary segments as a function of depth was presented in Suppl. Figure 7B in (Sakadzic et al., 2014), which are both directly proportional to the density of the capillary segments. It is correct that the capillary segment density is not the same as the vessel density, but we believe that capillary segment density is even more relevant for considerations of the relation between the mean capillary flow and the total flow (i.e. that considering the length of individual segments may be less important than the number of branching segments off the main supplying vessel). Nevertheless, in Figure D2 in (Blinder et al., 2013), average of the first two bars (layer I extends to ~100 µm depth in the mouse barrel cortex) is close to 50% smaller than the bars in the layer IV.

While we believe that measurements of the capillary RBC flux and segment density provide strong indication that blood flow in layer IV is larger than in the superficial layers, it will be important to validate this conclusion in the future by direct layer-dependent CBF measurements. Prompted by the reviewer’s comments, we expanded the Discussion section to better clarify that blood flow, oxygen extraction, and arterio-venous saturation difference are mutually dependent observables, and to point out that future direct measurements of CBF in different cortical layers will further strengthen our conclusions (please see the answer to summary comment #2 for details about the inserted text).

I believe the impact statement should be improved. "Homogenization", "Mechanism" and "adapts" suggest that the presented study looks at active mechanisms or dynamic changes. However, the work is a detailed description of the steady state flow and oxygen distribution.

We thank the reviewer for this suggestion. Please see the response to the summary comment #4.

[Editors’ note: further revisions were suggested before acceptance, as shown below.]

[…] To end the controversy, I propose the following:1) Subsection “Oxygen extraction fraction increases in the deeper cortical layers”: substitute the sentence by something like: "Note that as the instantaneous RBC shadow varies with both RBC speed, position and vessel size (see Supplementary Figure 1), RBC speed calculation was model-based by assuming a constant RBC size (6µ)(Unekawaet al., 2010).2) In Supplementary Figure 1 (it will replace the current supplementary figure 1 which is not informative) please add the following plots which are interesting, justify the model based-choice and explain the problem of point measurements to estimate the RBC speed:Add the plot from your Summary Comment (Title Comparison between the RBC speed measurements by the line-scan and point-scan method.)Add the new Author response image 7 plot (title Correlation between the line-scan and point-scan RBC-speed values in the capillaries having the diameter of 2-3 μm (left panel) and 3-5 μm (right panel)).Add the plot Author response image 1 from your former response (Title a-c. RBC longitudinal size vs. capillary diameter, RBC speed, and line-density, respectively.)3) Add the comments on these findings in the Discussion section.

Thank you for this suggestion. We updated the text in subsection “Oxygen extraction fraction increases in the deeper cortical layers”, which now reads:

“Please note that the RBC speed estimation in this work was model-based by assuming a constant RBC size (6 µm) (Unekawa et al., 2010). However, the RBC size may vary with RBC speed, line-density and capillary diameter (Chaigneau et al., 2003). The comparisons between the RBC speed measurements obtained by using the model-based point-scan method and more direct measurements by the line-scan method (Kleinfeld et al., 1998) are presented in Figure 1—figure supplement 2.”

To preserve the flow of this paragraph, this expanded text was added at the end of the first paragraph of sunsection “Oxygen extraction fraction increases in the deeper cortical layers” in the revised manuscript.

In the revised manuscript, the original Figure 1—figure supplement 1 was replaced and Discussion section was updated as suggested. Please note that Supplementary Figures 1 and 2 exchanged their positions due to their order of appearance in the manuscript text.

Smaller EATs in layer IV:Include in the discussion your detailed responses to the following points:The increase in interRBC-PO2 and RBC-PO2 over depth, which is a very surprising result, as generally the most saturated RBCs enter the vasculature at the surface.The impact of the sampling of "upstream" and "downstream" vessels and their distribution over depth.Saturation difference/oxygen extraction:As oxygen extraction, saturation difference and total blood flow are related quantities, it is important to add this information at two locations:-where the depth-dependent oxygen extraction fraction is introduced (subsection “Oxygen extraction fraction increases in the deeper cortical layers”) and-where the calculation of the depth-dependent OEF is described (subsection “Calculation of SO2 and depth-dependent OEF.”).

We accepted all suggestions in this comment and updated the manuscript text accordingly.